# Phytochemical Screening, Antibacterial, Antifungal, Antiviral, Cytotoxic, and Anti-Quorum-Sensing Properties of *Teucrium polium* L. Aerial Parts Methanolic Extract

**DOI:** 10.3390/plants9111418

**Published:** 2020-10-23

**Authors:** Mousa Alreshidi, Emira Noumi, Lamjed Bouslama, Ozgur Ceylan, Vajid N. Veettil, Mohd Adnan, Corina Danciu, Salem Elkahoui, Riadh Badraoui, Khalid A. Al-Motair, Mitesh Patel, Vincenzo De Feo, Mejdi Snoussi

**Affiliations:** 1Department of Biology, University of Hail, College of Science, P.O. Box 2440, Ha’il 2440, Saudi Arabia; mo.alreshidi@uoh.edu.sa (M.A.); eb.noumi@uoh.edu.sa (E.N.); n.vajid@uoh.edu.sa (V.N.V.); mo.adnan@uoh.edu.sa (M.A.); s.elkahoui@uoh.edu.sa (S.E.); ri.badraoui@uoh.edu.sa (R.B.); 2Laboratory of Bioressources, Integrative Biology and Recovery, High Institute of Biotechnology–University of Monastir, Monastir 5000, Tunisia; 3Laboratory of Bioactive Substances, Center of Biotechnology of Borj Cedria (CBBC), BP 901, Hammam Lif 2050, Tunisia; lamjed.bouslama@gmail.com; 4Ula Ali Kocman Vocational School, Mugla SitkiKocman University, Mugla 48147, Turkey; ozgceylan@hotmail.com; 5Department of Pharmacognosy, Faculty of Pharmacy, “Victor Babes” University of Medicine and Pharmacy, 2 Eftimie Murgu Square, Timisoara 300041, Romania; corina.danciu@umft.ro; 6Section of Histology—Cytology, Medicine College of Tunis, Tunis El Manar University, Road Djebel Lakhdhar, La Rabta-Tunis 1007, Tunisia; 7Laboratory of Histo-Embryology and Cytogenetic, Medicine College of Sfax, Sfax University, Sfax 3029, Tunisia; 8Molecular Diagnostic and Personalized Therapeutics Unit, University of Ha’il, Ha’il 2440, Saudi Arabia; mbkhm65@gmail.com; 9Bapalal Vaidya Botanical Research Centre, Department of Biosciences, Veer Narmad South Gujarat University, Surat, Gujarat 395007, India; patelmeet15@gmail.com; 10Department of Pharmacy, University of Salerno, Via Giovanni Paolo II, 132, Fisciano, 84084 Salerno, Italy; defeo@unisa.it; 11Laboratory of Genetics, Biodiversity and Valorisation of Bioresources, High Institute of Biotechnology–University of Monastir, Monastir 5000, Tunisia

**Keywords:** *Teucrium polium* L., aerial parts, bioactive compounds, HR-LCMS, biological activities

## Abstract

The chemical profile of *Teucrium polium* L. (*T. polium*) methanolic extract was tested using liquid chromatography coupled with high resolution mass spectrometry (HR-LCMS). Disc diffusion and microdilution assays were used for the antimicrobial activities. Coxsackievirus B-3 (CVB3) and Herpes simplex virus type 2 (HSV-2) were used for the antiviral activities. *Chromobacterium violaceum* (ATCC 12472 and CV026) and *Pseudomonas aeruginosa* PAO1 were used as starter strains for the anti-quorum sensing tests. Isoprenoids are the main class of compounds identified, and 13R-hydroxy-9E,11Z-octadecadienoic acid, valtratum, rhoifolin, sericetin diacetate, and dihydrosamidin were the dominant phytoconstituents. The highest mean diameter of growth inhibition zone was recorded for *Acinetobacter baumannii* (19.33 ± 1.15 mm). The minimal inhibitory concentrations were ranging from 6.25 to 25 mg/mL for bacterial strains, and from 6.25 to 25 mg/mL for *Candida* species. The 50% cytotoxic concentration on VERO (African Green Monkey Kidney) cell lines was estimated at 209 µg/mL. No antiviral activity was recorded. Additionally, *T. polium* extract was able to inhibit *P. aeruginosa* PAO1 motility in a concentration-dependent manner. However, the tested extract was able to inhibit 23.66% of the swarming and 35.25% of swimming capacities of PAO1 at 100 µg/mL. These results highlighted the role of germander as a potent antimicrobial agent that can interfere with the virulence factors controlled by the quorum-sensing systems.

## 1. Introduction

Aromatic and medicinal plants’ history is associated with the evolution of civilizations and occupies a crucial place in medicine, cosmetology, and culinary preparations in China, India, the Middle East, Egypt, Saudi Arabia, and Greece [1,2,3,4,5,6]. For thousands of years, humanity has used various plants found in its environment, to treat and cure all kinds of diseases [7,8,9,10]. Secondary metabolites still remain the object of much in vivo research as well as in vitro, including phenolic compounds, saponosides, alkaloids, and volatile oils [11,12].

Nowadays, infectious diseases are a major cause of human troubles around the world. As a result, control of microorganisms, especially multidrug-resistant pathogens, is increasingly important for the development of effective treatment against infectious diseases [13,14,15,16]. New approaches have been proposed to fight microbial resistance, such as nanotechnology [17], to use natural products due to their healing potential. This idea dates back to ancient times and has been taken up and developed in recent years, especially against multi-drug resistant human pathogenic microorganisms [18,19,20,21]. As a result, plants—especially aromatic/medicinal ones—appear to be a potential source of antimicrobial agents due to their richness in alkaloids, anthraquinones, saponins, terpenoids, tannins, and polyphenols [22,23].

In Saudi Arabian folk medicine, germander (*T. polium*) is known under the vernacular name of “Jaada” used by Bedouins as a febrifuge and vermifuge to fight stomach and intestinal troubles. The plant is also used in cold and fever [24,25,26]. In the Mediterranean basin, many scientific reports highlighted the medicinal benefits of germander [27,28,29,30]. Few studies have focused on the chemical oil of *T. polium* in Middle East [26,31]. Most of the research on *T. polium* tends towards the uses of the plant and plant- based extract as antimicrobial compounds. This plant species is considered as an important herb in folk medicine prescriptions. Since many years, this plant has been recommended as a medication for its anti-diabetic, anti-inflammatory, anti-spasmodic, and antioxidant effects [32,33,34].

Different techniques were used to identify the bioactive compounds in *T. polium* aerial parts extracts, using different solvents (water, methanol, 65.5% menthol–water, dichloromethane/methanol, acetone, ethanol 70%, and ethanol 95%) from different regions around the world [35,36,37,38,39,40,41,42,43,44]. A literature review of the phytochemical compounds isolated from *T. polium* aerial parts showed the identification of more than 130 molecules dominated by terpenoids (60%) [45]. The main identified compounds from *T. polium* roots, aerial parts, and inflorescence are apigenin, luteolin, rutin, cirsiliol, cirsimaritin, salvigenin, and eupatorin [45].

Five flavonoids were widely isolated from different *Teucrium* species from European countries, namely: cirsiliol, cirsimaritin, cirsilineol, salvigenin, and 5-hydroxy-6,7,3′,4′-tetramethoxyllavone [44]. Four new sesquiterpenes were identified in dichloromethane/methanol extract from *T. polium*, namely 4β,5α-epoxy-7αH-germacr-10(14)-en-6β-ol-1-one, 4β,5α-epoxy-7αH-germacr-10(14)-en,1β-hydroperoxyl,6β-ol, 4β,5β-epoxy-7αH-germacr-10(14)-en,1β-hydroperoxyl,6β-ol, and 4α,5β-epoxy-7αH-germacr-10(14)-e [39].

For this purpose, the main objectives of the present investigation were to assess for the first time the phytochemical profile of *T. polium* methanolic extract from the Hail region (Saudi Arabia) using HR-LCMS. The antimicrobial activities of germander extract against a wide range and types of clinical bacterial and fungal isolates were also tested. The antiviral activity was tested for the first time against CVB3 and HSV-2. Inhibition of the production of some virulence factors in *C. violaceum* and *P. aeruginosa* PAO1 reporter strains were also investigated for the first time in the present work.

## 2. Results

### 2.1. Phytochemical Composition

Our results showed the identification of 29 molecules in *T. polium* methanolic extract using the HR-LCMS technique. The complete list of the identified compounds together with their respective chemical class and their chemical structure are listed in Table 1.

In fat, *T. polium* methanolic extract was dominated by terpenes and their derivatives, fatty acids and their derivatives, polyphenols, flavonoids and derivatives, coumarines, amino acid derivatives, and alkaloids. The tested extract was dominated by 3R-hydroxy-9E,11Z-octadecadienoic acid, rhoifolin, sericetin diacetate, dihydrosamidin, and cepharanthine. The chromatograms (Figure 1) showed respectively major peaks indicating the presence of various phytochemical constituents. Data related to MS spectra, MS/MS spectrum peak list, chemical formula, and m/z values are listed in the Appendix A.

The chemical structure of the identified 29 compounds are listed in Figure 2.

### 2.2. Cytotoxicity Assay

Results showed that at the 50% cytotoxic concentration (CC_50_), defined as the concentration of the extract reduced by 50%, the VERO cell viability was estimated at 209 µg/mL (Figure 3).

### 2.3. Antimicrobial Activities

Table 2 illustrated the results of antibacterial activities of *T. polium* aerial parts methanolic extract against nine Gram-negative and three Gram-positive bacteria using disc diffusion assay. The results revealed that the methanolic extract exhibited a wide antibacterial spectrum, against tested bacteria with the inhibition zone diameter fluctuant from 6 to 20 mm. Our results showed various degrees of the inhibition. In fact, very high antibacterial activity was recorded against the clinical isolates *A. baumannii* (strain 146) and *S. pyogens*. A high activity level was recorded against *P. aeruginosa* ATCC 27853, *P. mirabilis* ATCC 29245, *P. aeruginosa* (environmental strain, pf8), and *E. cloacae* (clinical strain, 115). Moderate activity was obtained against *E. coli* ATCC 35218, *K. pneumoniae* ATCC 27736, *S. aureus* MDR (clinical strain, 136), and *S. paucimobilis* (clinical strain, 144). Low activity was recorded against *P. mirabilis* (environmental strain, 3), and *S. sciuri* (environmental strain). The results of the minimal bactericidal concentration/minimum inhibitory concentration (MBC/MIC) ratios interpreted using the scheme of antimicrobial substances showed that *T. polium* aerial parts methanolic extract acts like a bacteriostatic agent for all tested bacteria except against *E. cloacae*, against which it is bactericidal.

The methanolic extract of *T. polium* showed varied antifungal activity (Table 3). Our results indicate that the extract exhibited a moderate inhibitory effect against unicellular yeast (*C. albicans*, ATCC 10231, *C. vaginalis*, and *C. neoformans* ATCC 14116) and low activity against filamentous fungi (*A. fumigatus* ATCC 204305 and *A. niger*). By comparing the results obtained in the presence of Amphotericin B, we observe that only for *C. vaginalis* (clinical strain, 136) the diameter of inhibition zone is higher that the used control. Based on the minimum fungicidal concentration (MFC)/MIC ratio, antifungal compounds present in our extract acts as fungicidal agents against the tested fungi.

### 2.4. Antiviral Activities

Results showed that the tested *T. polium* methanolic extract was not active against the two viruses used: CVB3 and HSV-2.

### 2.5. Anti-Quorum-Sensing Activities

The anti-quorum-sensing activities were tested using two starter strains (*C. violaceum* wild and mutant strains) and *P. aeruginosa* PAO1. The results showed that *T. polium* methanolic extract has no effect on the violacein production in the microtiter plate assay using the mutant strain *C. violaceum* CV026 at 5 mg/mL (Figure 4A). Additionally, no inhibition of the violacein pigmentation produced by the wild type, *C. violaceum* ATCC 12472, on Luria–Bertani Petri dishes of agar was observed (Figure 4B).

Table 4 shows the effect of different concentration of *T. polium* methanolic extract on two virulence properties controlled by the quorum-sensing system in *P. aeruginosa* species: swarming and swimming features.

Results demonstrated that this plant extract was able to inhibit the swarming and swimming activities in a concentration-dependent manner. The highest inhibition was recorded at 100 µg/mL with a percentage of motility inhibition about (35.25 ± 2.5%) for swimming activity, and (23.66 ± 0.5%) for swarming activity.

## 3. Discussion

The yield of extraction using pure methanol as solvent was about 9.437 ± 0.114%. Similar results were obtained using the same solvent and the aerial parts of felty germander—the yield of extraction was 8.24% [46]. Previous studies reported different yields of extraction using organic solvents with different polarities, and different *T. polium* organs [47,48,49,50]. In fact, using ethanol (50%) and the *T. polium* aerial parts, the yield of extraction was 12.5% [48]. The yields of extraction from aqueous and ethanol extracts of *T. polium* L. subsp. *gabesianum* (L.H.) aerial parts were 9% and 7%, respectively [50].

Additionally, Alzeer et al. [49] used various solvents, and the yields of extraction were estimated at 17% (90% ethanol), 20% (80% methanol), 7.4% (acetone), 23% acetic acid, 25% (apple vinegar), 9% (grape vinegar), and 53% (coconut water). Using boiling water (1:10, *w/v*) and ethanol 96% (1:8, *w/v*) to elute bioactive compounds from *T. polium* aerial parts, the yields of extraction were 16.3% and 14.7%, respectively [48].

Nematollahi-Mahani et al. [51] studied the toxicity of *T. polium* 96%-ethanolic extract on different known malignant cell lines like A549 (human lung adenocarcinoma), BT20 (human breast ductal carcinoma), MCF-7 (human breast adenocarcinoma), and PC12 (mouse pheochromocytoma). The results reported showed that IC50 values were 90 µg/mL (A549), 106 µg/mL (BT20), 140 µg/mL (MCF-7) and 120 µg/mL (PC12). The cytotoxic properties of *T. polium* aerial parts can be attributed the presence in the organic extracts of some diterpenoids and their acyl derivatives (teucvin and teucvidin), flavonoids (cirsiliol, cirsimaritin, cirsilineol, salvigenin, and 5-hydroxy-6,7,3′,4′-tetramethoxyllavone), saponin poliusaposide [52], and selenium [53,54,55].

In a systematic review, Harborne et al. [44] reported the distribution of free flavone aglycones and flavonoid glycosides from 42 European taxa of the genus *Teucrium* (75% ethanol extract from aerial parts). These authors reported the identification of 5,7-Dihydroxytkvones, 6-hydroxyBavones, 6-methoxytIavoncs, flavonols, 8-hydroxyflavones, cirsiliol, cirsimaritin, citsilineol, 5-hydroxy-6,7,3′,4′-tetramethoxyllavone, salvigenin, luteolin, apigenin, diosmetiq, hydroxyluteolin, cirsimariti, quercetin, isorhamnetin, hypolaetin, isoscutellarein, vicenin-2, 5,7dihydroxytlavone glycosides, 6-hydroxytIavone glycosides, I-hydroxyflavone glycoside, 6-methoxyflavone glycosides, tlavonol glycoside, apigenin-7-glucoside, apigenin 7-rutinoside, luteolin 7-glucoside, lutcolin 7-rutinoside, luteolin 7-sambubiosidc, diosmetin 7-rutinosidc, 6-OH-luteolin 7-glucoside, 6-OH-luteolin 7-rhamnoside, isoscutellarein 7-allosylglucoside, hypolaetin 7-allosylglucoside, cirsimaritin 4′-glucoside, quercetin 3-glucoside, quercetin 3-rutinoside, isorhanmetin 3-glucoside, and isorhamnetin 3-rutinoside.

Table 5 summarizes some important bioactive molecules isolated from the *Teucrium* species, mainly *T. polium*, using various identification techniques and different extraction procedures [35,36,37,38,39,40,41,42,43,44]. 

In the present work, we reported the identification of a new undescribed compounds in *T. polium* aerial parts methanolic extract using the HR-LCMS technique: 13R-hydroxy-9E,11Z-octadecadienoic acid, rhoifolin, sericetin diacetate, selinidin, harpagoside, valtratum, triptonide, koparin 2′-methyl ether, dihydrosamidin, 10S,11R-epoxy-punaglandin, 4, 16alpha, 17beta-Estriol 3-(beta-D-glucuronide), khayanthone, 10-hydroxyloganin, 7-epiloganin tetraacetate, cepharanthine, deoxyloganin tetraacetate, carapin-8 (9)-Ene, and 1-dodecanoyl-sn-glycerol. 

In previous work, Pacifico et al. [42] reported the identification of poliumoside, apigenin, luteolin, montanin D, montanin E, teubutilin A, teuchamecrin C, teulamifin B, teupolin VI, teupolin VII, teupolin VIII, teupolin IX, teupolin X, teupolin XI, and teupolin XII from T. polium methanolic extract from Italy. Using the same solvent, De Marino et al. [40] identified poliumoside B, poliumoside, teucardosid, 8-O-acetylharpagid, luteolin7-O-rutinoside, luteolin 7-O-neohesperidoside, luteolin 7-O-glucosied, luteolin 4′-O-glucoside, teulamifin, and teusalvin C as main bioactive compounds from *T. polium* aerial parts from Italy. Meanwhile, Sharififar et al. [43] identified 3′,6 dimethoxy apigenin, rutin, apigenin, and 4′,7 dimethoxy apigenin as the main bioactive molecules in *T. polium* methanolic extract from Iran.

Previous reports have underlined that methanolic extract from *T. polium* showed biological activity against Gram-negative and Gram-positive bacteria [56]. Mainly the antibacterial activity was exhibited against Gram-negative bacteria. The antibacterial activity of aerial parts’ methanolic extract of *T. polium* is directly linked to their phytoconstituents and contingent by the activity of their chemical compounds and their proportion in the extract. Many studies evaluated the antibacterial activity of the essential oil of *T. polium* [29,50]. The essential oils, aqueous, and ethanol extracts of *T. polium* exhibited high antibacterial activity, against both Gram-negative and positive bacteria and antifungal properties [50]. Recently, El Atki et al. [31] examined the antibacterial activities of the essential oils of two Moroccan *T. polium* L. subspecies. Moreover, Ghojavand et al. [57], synthesized green nanoparticles from the stem and flower of *T. polium* and investigated their antifungal activity. In fact, previous studies demonstrated that methanolic extract exhibited high antibacterial activity [58,59,60,61]. Comparing the efficacy of our extract to ampicillin, we found that it is more highly active than a commonly used antibiotic against *A. baumannii*, *E. cloacae*, *P. aeruginosa*, *P. mirabilis* ATCC 29245, and *P. aeruginosa* ATCC 27853. Previous study conducted by Mosadegh et al. [62] demonstrated that aqueous extract from *T. polium* exhibited antibacterial activity but not any antifungal properties. Tabatabaei et al. [63] demonstrated that *T. polium* extract showed antibacterial properties against infectious microorganisms in vitro, notably against Gram-positive strains.

For the last several years, *T. polium* has been tested and was considered to be a plant rich in antibacterial compounds [56]. In fact, it exerts antibacterial activities against Gram-negative and positive bacteria [54]. Many research groups [59,64,65,66] obtained the same result and demonstrated that the aspect, constituents, and architecture of the membranes of Gram-negative bacteria makes them more susceptible to the bioactive compounds present in methanolic extract. This finding could be explained by their hydrophilic outer membranes, composed of a single layer of peptidoglycan thatinvariably contains a unique component; lipopolysaccharide, which, in addition to proteins and phospholipids, makes the membranes more permeable for any antibacterial compound [31]. Therefore, the Gram-negative strains are more sensitive than Gram-positive bacteria. The antibacterial effect of the *Teucrium* species has been widely demonstrated [29,67]. In fact, Stanković et al. [67] studied the antimicrobial activity of seven taxa of the genus *Teucrium* and they used several solvents (21 extracts). The result indicated that the plant extract showed greater potential for antibacterial than antifungal activity. Darabpour et al. [50] demonstrated that the *E. coli* and *S. aureus* species were sensitive against *T. polium* extract. The S. *aureus* MDR was found to be one of the most resistant microorganisms against *T. polium* leaves methanol extract [68].

The microdilution method is more sensitive and permits the quantification of extract biological activity by the determination of MICs and MBCs/MFCs concentrations. Regarding MIC values, many studies confirm that, if these values are below 8 mg/mL, the extract is considered to have antimicrobial properties [69]. Moreover, if this value is lower than 2 mg/mL the extract is considered strongly active [70,71]. The MIC result of methanolic extract against *E. cloacae* (clinical strain, 115) was 25 mg/mL. However, it was 6.25 mg/mL against *E. coli* ATCC 35218, *P. mirabilis* ATCC 29245, *P. mirabilis* (environmental strain 3), *S. pyogens* (clinical strain), and *P. aeruginosa* (environmental strain, pf8). Moreover, it was 12.5 mg/mL for the rest of the tested strains. The result of the MBC/MIC ratio interpreted using the scheme of antimicrobial substances shows that *T. polium* aerial parts methanolic extract acts like a bacteriostatic agent for all tested bacteria, except against *E. cloacae* where it acts as a bactericidal agent. This result was in agreement with other studies which demonstrated that methanolic extract inhibits bacterial growth and does not kill them [60,72].

Several solvents have been used to extract and evaluate the antibacterial activity of the aerial part of *T. polium*. Indeed, organic (ethanol, methanol, acetone, ethyl acetate, and chloroform) or aqueous [64,67,73] extracts were used to investigate the antibacterial activity of *T. polium*. The result indicated that the organic extract showed greater activity than the aqueous extract, which defends our choice in the use of methanol for the extraction of active compounds. Other study shows that ethanol extract of *T. polium* exhibited antibacterial activities against several bacteria among them are antibiotic resistant strains [74]. It was also demonstrated that *T. polium* had antifungal activity against fungal species (*C. albicans* and *A. niger*). Furthermore, the results of Kremer et al. [75] investigating especially *T. arduini* L. (Lamiaceae), showed that the plant had the antimicrobial properties against *S. aureus*, *C. albicans*, and various other microorganisms.

Similarly, Taheri et al. [76] studied the effect of extracts from *A. sieberi*, which showed antifungal activity against *C. albicans*, and prevented its growth. The results support the findings of our work. Indeed, the plant extract stopped the growth of *C. albicans*. Additionally, Gholampour-Azizi et al. [77], studied the in vitro antifungal activity of *Cucumis melo* and he demonstrated that it exhibited specific antifungal activity against *C. albicans*. Likewise, it has been demonstrated that aqueous and ethanolic extracts from *T. polium* showed an antifungal effect against *C. albicans* [78]. Furthermore, another study demonstrated that the aqueous extract of *T. polium* (2.5, 10, and 20 g/L) slows down the growth of *Saccharomyces cerevisiae* and *Yarrowia lipolytica*. In a recent study, Akbarzdeh et al. [79] studied the sensitivity of *Candida* strains against a *T. polium* smoke product. The authors demonstrated that all the standard clinical samples of *Candida* were sensitive. On the other hand, the result of the study of the antimicrobial activity of acetone and chloroform extracts from *T. polium* against four fungal species and eleven bacteria showed that the antibacterial effect is more significant than the antifungal activity, which is in concordance with our results [67,73].

Additionally, previous results using aqueous and ethanolic extracts from *T. polium* aerial parts have shown moderate anti-rinderpest virus [80]. In 2014, Ansari and co-workers [81] reported that *T. polium* was a rich source of rosmarinic acid (1.8%, *w/w*), weak cytotoxic activity (maximum non-toxic concentration = 1000 μg/mL), and anti-Herpes Simplex Virus-1 (HSV-1)—affected by the exposure time and the concentration of the extract used. In fact, after one hour of cell infection, 93.1% of HSV-1 were inhibited by 50 μg/mL of plant extract. This percentage of inhibition reached 100% after one hour of exposure to 100 and 250 μg/mL of felty germander extract [81]. This activity was correlated to the high concentration of rosmarinic acid, a phenolic compound known to possess antiviral activity [82].

Many plant extracts and essential oils have been reported to possess high anti-quorum-sensing potential and play an important role in attenuating the secretions of many virulence factors in *P. aeruginosa* PAO1 and *C. violaceum* reporter strains [83,84,85,86,87]. According to the literature review, only one study has reported the effect of *T. polium* alcohol extract against the production of the purple pigment by *C. violaceum* ATCC 12472, a phenomenon controlled by the quorum-sensing system. Results obtained reported that the tested extract had not showed a clearly visible white halo in the violacein bioassay [88]. It has been demonstrated that a 95% ethanolic extract of *T. polium* leaves was able to reduce the biofilm formed by *P. aeruginosa* strain on a polystyrene 96-well plate at 25 ppm [89]. More recently, a crude extract from air-dried aerial parts of felty germander from Egypt was able to decrease the biofilm formation by *S. aureus* strains at low concentration [90].

Overall, the biological properties of germander outlined in this study could certainly be the result of the phytochemical compounds present in the methanolic extract tested. In fact, several bioactive compounds have been reported in the organic extracts of the tested plant species [91]. Previous reports showed that *T. polium* possessed potential therapeutic properties in some health problems including toxicity and cytogenic diseases [92]. In this study, further efficient activities could be outlined particularly the ability to treat bacterial infectious diseases and to inhibit cell-to-cell communication (quorum sensing).

## 4. Materials and Methods

### 4.1. Plant Material Sampling and Extract Preparation

The plant material was collected in October 2019 from a nursery belonging to the Ministry of Agriculture, Hail, Saudi Arabia. A voucher specimen (AN01) was deposited at the herbarium in the Department of Biology, College of Science, University of Hail, Kingdom of Saudi Arabia. The fresh aerial flowering parts were dried at room temperature for seven to ten days. For the bioactive compound’s extraction, 4 g of the plant powder material were macerated in 40 mL absolute methanol (ratio: 1:10, *w/v*). Methanolic extracts were pooled, filtered, and the solvent was removed at 60 °C in the incubator chamber. The yield of extraction was about 9.437 ± 0.114%.

### 4.2. Identification of Bioactive Molecules by Liquid Chromatography Coupled with High Resolution Mass Spectrometry

The HR-LCMS analysis of *T. polium* was analyzed by using a UHPLC-PDA-Detector Mass spectrometer (HR-LCMS 1290 Infinity UHPLC System, 1260 Infinity Nano HPLC with Chipcube, 6550 iFunnel QTOFs), Agilent Technologies, USA. For chromatographic separation, an Agilent 1200 Series HPLC system (Agilent Technologies, Santa Clara, CA, USA) was used, equipped with a binary gradient solvent pump, HiP Sampler, column oven, and MS Q-TOF with Dual AJS ES Ion Source. Samples were separated on an SB-C18 column (2.1 × 50 mm, 1.8-particle size; Agilent Technologies, Santa Clara, CA, USA), maintained at 25 °C. The solvents used were: water containing 0.1% HCOOH, and acetonitrile containing 10% water and 0.1% HCOOH. The following gradient elution program at a flow rate of 0.3 mL min^−1^ was applied. MS detection was performed in an MS Q-TOF Mass spectrometer (Agilent Technologies). Compounds were identified via their mass spectra and their unique mass fragmentation patterns. Compound Discoverer 2.1, ChemSpider, and PubChem were used as the main tools for the identification of the phytochemical constituents [93].

### 4.3. Evaluation of the Cytotoxicity

The cytotoxicity of the *T. polium* extract was evaluated on VERO (African green monkey kidney) cell lines using the MTT colorimetric method. A decreasing concentration of the extract (from 3333 µg/mL to 1 µg/mL) diluted in MEM supplemented with 2%-FBS was applied on VERO cells in a 96-well plate. Cell controls were incubated under the same conditions without extracts. After 72 h of incubation at 37 °C, the extract dilutions were substituted with the MTT solution (5 mg/mL) and incubated for 2 h at 37 °C and then dissolved by the dimethyl sulfoxide (DMSO). The plate was read on an ELISA reader at a 570 nm wavelength to measure the optical density. The 50% cytotoxic concentration (CC_50_) was determined by regression analysis in comparison with a negative control [83]. 

### 4.4. Antimicrobial Activities

#### 4.4.1. Selected Microorganisms

The obtained methanolic extract was tested against a wide range of selected clinical/environmental microorganisms: *Escherichia coli* ATCC 35218, *Pseudomonas aeruginosa* ATCC 27853, *Proteus mirabilis* ATCC 29245, *Klebsiella pneumoniae* ATCC 27736, *Proteus mirabilis*, *Staphylococcus sciuri*, *Streptococcus pyogens*, *Pseudomonas aeruginosa*, *Staphylococcus aureus* MDR, *Enterobacter cloacae*, *Sphingomonas paucimobilis*, and *Acinetobacter baumannii*. Four yeast strains including *Candida albicans* ATCC 10231, *Cryptococcus neoformans* ATCC 14116, *Candida vaginalis*, and *Candida albicans* and two fungal strains; *Aspergillus fumigatus* ATCC 204305, and *Aspergillus niger* were used.

#### 4.4.2. Antibacterial and Antifungal Activity Using Disc Diffusion Assay

The disc diffusion assay was performed on Mueller–Hinton agar plates for all bacteria, Sabouraud chloramphenicol agar for yeasts, and potato dextrose agar for *Aspergillus* strains as previously described by Snoussi et al. [94].10 μL of methanolic extract mother solution (300 mg/mL) was used to impregnate 6 mm sterile discs. The experiment was done in triplicate. All Petri dishes were incubated overnight at 37 °C and the diameter of the growth inhibition zone was recorded using a 1 cm flat ruler. The mean diameter was recorded and expressed as (GIZ mm ± SD). Results obtained are interpreted using the scheme proposed by Parveen et al. [95]. No activity (GIZ 0), low activity (GIZ: 1–6 mm), moderate activity (GIZ: 7–10 mm), high activity (GIZ: 11–15 mm), and very high activity (GIZ: 16–20 mm). Ampicillin and Amphotericin B were used as control.

#### 4.4.3. MICs and MBCs/MFCs Determination Using the Microdilution Assay

MICs and MBCs/MFCs values were determined by a microtiter broth dilution method as previously described by Snoussi et al. [96]. The test medium was Mueller–Hinton Broth (MHB) for bacterial strains, and Sabouraud dextrose broth for fungal isolates. Bacterial/fungal suspensions (100 μL) were inoculated into the wells of 96-well microtiter plates in the presence of samples with different final concentrations (ranging from 0.039 mg/mL to 100 mg/mL). The inoculated microplates were incubated at 37 °C for 24 h. The lowest concentration of *T. polium* methanolic extract, which did not show any visual growth of tested organisms, was recorded as the MIC value (expressed in mg/mL). The MBCs/MFCs values are determined by streaking all wells after the MICs values on the correspondent agar media of the tested microorganisms. MBC/MIC ratio and MFC/MIC ratio were used to interpret the activity of the essential oil as described by Gatsing et al. [97].

#### 4.4.4. Antiviral Activities

The antiviral activity of *T. polium* methanolic extract was evaluated on Coxsackievirus B-3 (CVB3) and HSV-2. The viruses were propagated on permissive VERO cells in MEM medium supplemented with 2% FBS at 37°C under humidified 5% CO_2_ atmosphere. The infectious titer of the stock solution was 2 × 10^7^ TCID50/mL (50% tissue culture infectious doses/mL) for CVB-3 and 8 × 10^5^ PFU/mL (plaque formation units/mL, PFU) for HSV-2. A multiplicity of infection about 0.01 for CVB-3 and 0.1 for HSV-2 was seeded into cells monolayers cultivated in 96-well culture plates (2 × 10^4^ cells/wells), with different concentrations of the extract starting from the CC_50_ value. Plates without virus or extract were used as negative and positive controls, respectively. After 48 h incubation, the inhibition of the cytopathic effect (CPE) for CVB-3 and the plaque virus for HSV-2 was observed in an inverted microscope [94].

### 4.5. Anti-Quorum Sensing Activity

#### 4.5.1. Bioassay for Quorum-Sensing Inhibitory (QSI) Activity Using Chromobacterium Violaceum CV026

The ability of the tested extract to inhibit the production of violacein by *C. violaceum* strains (ATCC 14272 and CV026) was assayed using the methods described by Noumi et al. [86]. Serial dilutions of the sample (MIC, MIC/2, MIC/4, MIC/8, and using LB broth as the diluent) were tested against *C. violaceum* CV 026 and the assay plates were incubated at 30 °C for 3 days. Each sample was tested in triplicate. Inhibition of quorum sensing was calculated using the protocol described by Zaki et al. [98] and activity is interpreted using the following scheme: a quorum sensing inhibition zone < 10 mm was considered moderate activity and when the QSI zone > 10 mm it was designated a potent effect.

#### 4.5.2. Violacein Inhibition Assays Using Chromobacterium Violaceum CV12472

Violacein inhibition was also tested on *C. violaceum* ATCC 12472 as previously described by Noumi et al. [86]. Overnight culture (10 μL) of the starter strain adjusted to 0.4 OD at 600 nm was used to inoculate sterile microtiter plates containing 200 μL of Luria–Bertani broth and incubated in the presence and absence of various concentrations of tested germander extract (MIC-MIC/16). All 96-well plates were incubated at 30 °C for 24 h and observed for the reduction in violacein pigment production. Each experiment was performed in triplicate. The absorbance was read at 585 nm. The percentage of violacein inhibition was calculated by following formula (Equation (1)):Violacein inhibition (%) = [(OD_control_ − OD_sample_)/OD_control_] × 100(1)

#### 4.5.3. Swarming and Swimming Motility Assays Using Pseudomonas Aeruginosa PA01

The effects of increasing concentrations of the methanolic germander extract (50, 75, and 100 μg/mL) were tested against the swarming/swimming motility of *P. aeruginosa* PAO1 strain was tested as previously reported by Noumi et al. [85]. For the swimming activity, the optical density of an overnight culture of PAO1 strain was adjusted at 0.4 OD at 600 nm and point inoculated at the center of 0.3% agar medium. For swarming assays, 5 μL of the same PA01-adjusted culture were inoculated at the center of the 0.5% agar medium. Then, the inoculated Petri dishes were incubated at 30 °C in upright position for 16 h. The reduction in swimming and swarming migration was recorded by measuring the swim and swarm zones of the bacterial cells after 16 h. Each sample was tested in triplicate and the mean zone of inhibition was recorded.

### 4.6. Statistical Analysis

All measurements were carried out in triplicate and the results were presented as mean values ± SD (standard deviations). ANOVA and Duncan tests were performed using SPSS 16.0. The means of the test values were also evaluated with the Least Significant Differences test at the 0.05 significance level.

## 5. Conclusions

Taken together, the obtained results showed that *T. polium* methanolic extract is a rich source of undescribed bioactive molecules in the *Teucrium* genus obtained using the HR-LCMS. Our findings highlighted the antimicrobial activities of *T. polium* methanolic extract from the Hail region, Saudi Arabia, against a large collection of clinically/environmentally important pathogens. *T. polium* extract was able to inhibit the motility of *P. aeruginosa* PAO1, reducing its virulence. These findings support the use of this plant in folk medicine as a source of bioactive molecules acting on several human disorders. Further analyses are needed to correlate between the obtained results with the phytochemical composition of organic extract from germander aerial organs.

## Figures and Tables

**Figure 1 plants-09-01418-f001:**
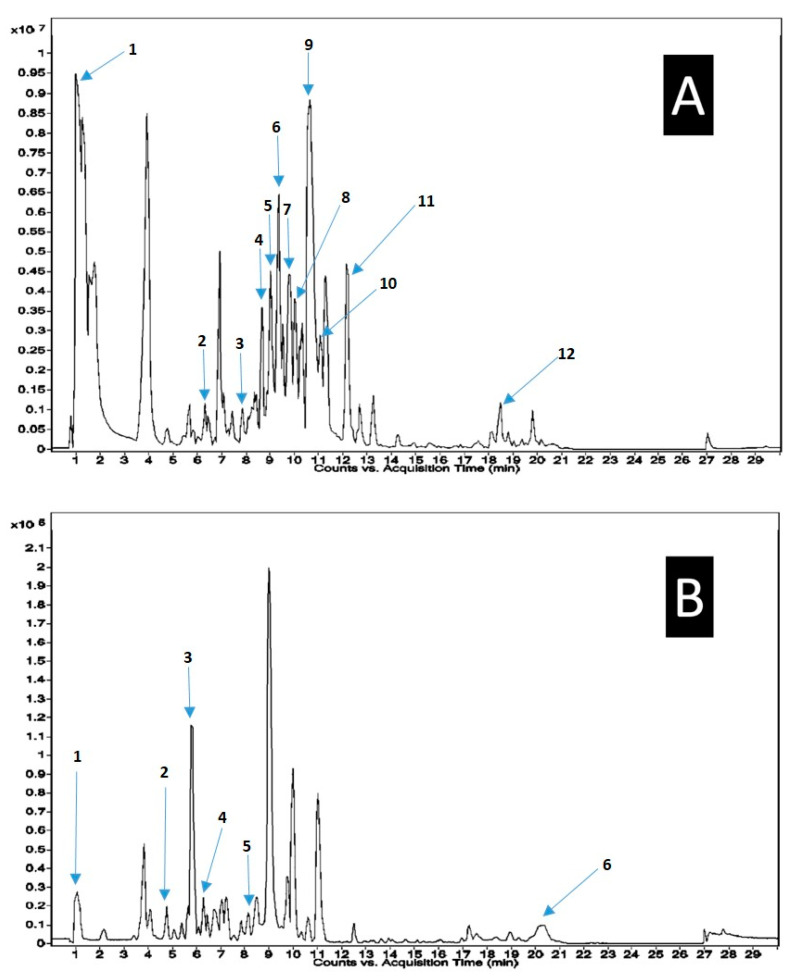
Main identified phytocompounds in *T. polium* methanolic extract using the HR-LCMS technique. (**A**): +ve Chromatogram—(1). 13R-hydroxy-9E,11Z-octadecadienoic acid, (2). Rhoifolin, (3). Sericetin diacetate, (4). Selinidin, (5). Harpagoside, (6). Valtratum, (7). Triptonide, (8). Koparin 2′-Methyl Ether, (9). Dihydrosamidin, (10). 10S,11R-Epoxy-punaglandin, (11). 4, 16alpha, 17beta-Estriol 3-(beta-D-glucuronide), (12). Khayanthone. (**B**): -ve Chromatogram—(1). 10-Hydroxyloganin, (2). 7-Epiloganin tetraacetate, (3). Cepharanthine, (4). Deoxyloganin tetraacetate, (5). Carapin-8 (9)-Ene, (6). 1-dodecanoyl-sn-glycerol.

**Figure 2 plants-09-01418-f002:**
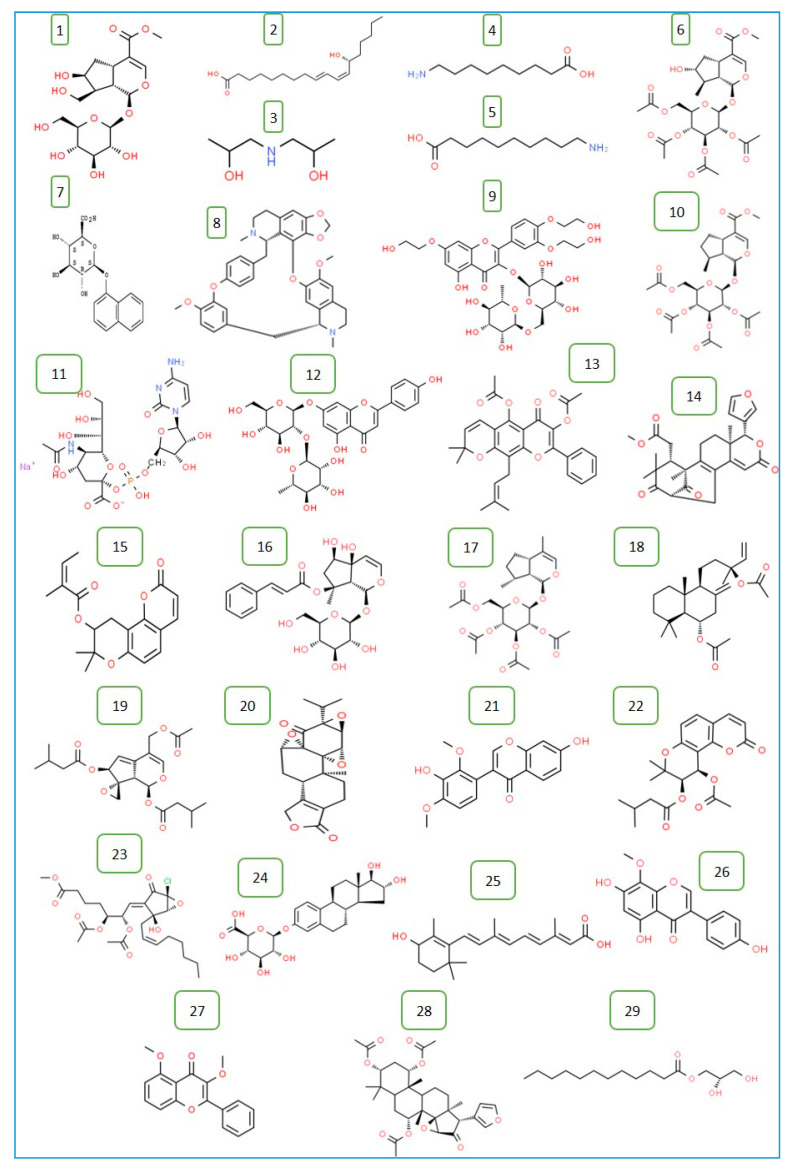
Chemical structure of the 29 compounds identified in *T. polium* methanolic extract by HR-LCMS. The names of compounds (1–29) are listed in Table 1.

**Figure 3 plants-09-01418-f003:**
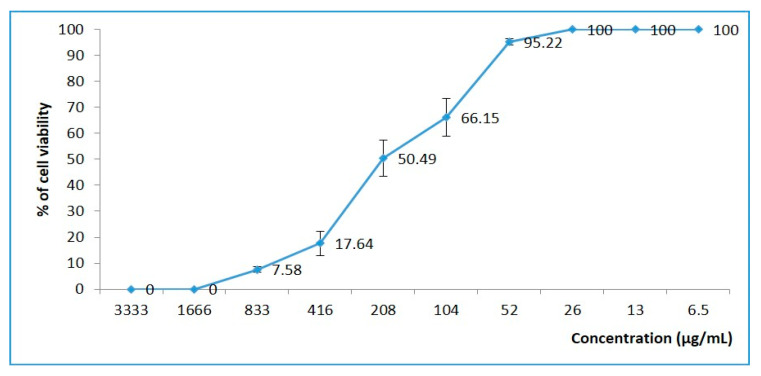
Cytotoxicity evaluation of *T. polium* methanolic extract against VERO cell lines using the MTT colorimetric method. Cell culture without extracts was used as negative control. The data represent the percentage of cell viability in comparison with the cell control, and the bars denote standard deviation (SD).

**Figure 4 plants-09-01418-f004:**
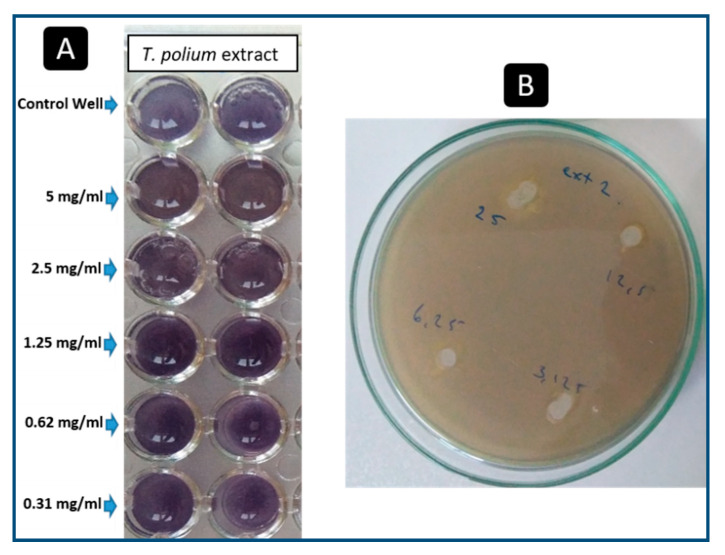
Determination of the effect of different concentrations of *T. polium* methanolic extract on violacein production by *C. violaceum* CV026 (**A**), and *C. violaceum* ATCC 14272 (**B**).

**Table 1 plants-09-01418-t001:** Bioactive compounds and their chemical class identified in *T. polium* methanolic extract using the HR-LCMS technique.

N°	RT	Identified Compound Name	Chemical Class	Chemical Formula
**1**	0.945	10-Hydroxyloganin	Isoprenoid	C_17_H_26_O_11_
**2**	1.046	13R-hydroxy-9E,11Z octadecadienoic acid	Octadecanoid	C_18_H_32_O_3_
**3**	1.062	Bis (2-hydroxypropyl) amine	Amino alcohol	C_6_H_15_NO_2_
**4**	1.447	9-amino-nonanoic acid	Amino fatty acid	C_9_H_19_NO_2_
**5**	3.807	10-Amino-decanoic acid	Amino fatty acid	C_10_H_21_NO_2_
**6**	4.739	7-Epiloganin tetraacetate	Isoprenoid	C_25_H_34_O_14_
**7**	5.342	b-D-Glucopyranoside uronic acid, 6-(3-oxobutyl)- 2-naphthalenyl	Organic acid, Phenol	C_20_H_22_O_8_
**8**	5.948	Cepharanthine	Alkaloid	C_37_H_38_N_2_O_6_
**9**	5.96	Troxerutin	Flavonol	C_27_H_30_O_14_
**10**	6.319	Deoxyloganin tetraacetate	Isoprenoid	C_29_H_28_O_7_
**11**	6.329	CMP-N-acetylneuraminic acid	Amino sugar	C_33_H_42_O_19_
**12**	6.376	Rhoifolin	Flavonoid	C_25_H_34_O_13_
**13**	7.838	Sericetin diacetate	Flavonol	C_20_H_31_N_4_O_16_P
**14**	8.198	Carapin-8 (9)-Ene	Limonoid	C_27_H_30_O_7_
**15**	8.848	Selinidin	Coumarin derivative	C_19_H_20_O_5_
**16**	9.1	Harpagoside	Iridoid glycoside	C_24_H_30_O_11_
**17**	9.126	8-Epiiridodial glucoside tetraacetate	Isoprenoid	C_24_H_34_O_11_
**18**	9.262	Larixol Acetate	Bicyclic labdane diterpenes	C_22_H_36_O_3_
**19**	9.525	Valtratum	Terpene	C_22_H_30_O_8_
**20**	9.807	Triptonide	Diterpene triepoxide	C_20_H_22_O_6_
**21**	10.036	Koparin 2′-Methyl Ether	Isoflavonoid	C_17_H_14_O6
**22**	10.779	Dihydrosamidin	Coumarins	C_21_H_24_O_7_
**23**	11.022	10S,11R-Epoxy-punaglandin 4	Eicosanoid	C_25_H_35_ClO_9_
**24**	11.279	16alpha,17beta-Estriol 3-(beta-D- glucuronide)	Steroidal glycosides	C_24_H_32_O_9_
**25**	11.28	16-Hydroxy-4-carboxyretinoic Acid	Isoprenoid	C_20_H_24_O_5_
**26**	12.149	Isotectorigenin, 7- Methyl ether	Isoflavonoid	C_18_H_16_O_6_
**27**	13.274	3-hydroxy-3′,4′- Dimethoxyflavone	Flavonoid	C_17_H_14_O_5_
**28**	18.427	Khayanthone	Limonoid	C_32_H_42_O_9_
**29**	20.37	1-dodecanoyl-sn-glycerol	Glycerolipid	C_14_H_22_N_2_O_3_

**Table 2 plants-09-01418-t002:** Growth inhibition zone, minimum inhibitory concentration (MIC) and minimal bactericidal concentration (MBC) values obtained for bacterial strains tested using disc diffusion and microdilution assays.

Code	Strains/Origin	*T. polium* Methanolic Extract				Ampicillin
		Mean ± SD * (mm)	MIC ^a^	MBC ^b^	MBC/MIC Ratio	Mean ± SD (mm)
**B_1_**	*Escherichia coli* ATCC 35218	7.00 ± 0.00 *^a^*	6.25	>100	>16	7.00 ± 0.00 *^a^*
**B_2_**	*Pseudomonas aeruginosa* ATCC 27853	12.33 ± 0.57 *^b^*	12.5	>100	>8	7.33 ± 0.57 *^a^*
**B_3_**	*Proteus mirabilis* ATCC 29245	11.66 ± 0.57 *^b^*	6.25	>100	>16	6.33 ± 0.57 *^a^*
**B_4_**	*Klebsiella pneumoniae* ATCC 27736	7.00 ± 0.00 *^a^*	12.5	>100	>8	6.66 ± 0.57 *^a^*
**B_5_**	*Proteus mirabilis* (environmental strain, 3)	6.00 ± 0.00 *^a^*	6.25	>100	>16	21.00 ± 1.00 *^d^*
**B_6_**	*Staphylococcus sciuri* (environmental strain, 4)	6.00 ± 0.00 *^a^*	12.5	>100	>8	7.00 ± 0.00 *^a^*
**B_7_**	*Streptococcus pyogens* (clinical strain)	16.66 ± 1.15 *^d^*	6.25	50	>16	16.00 ± 1.73 *^c^*
**B_8_**	*Pseudomonas aeruginosa* (environmental strain) pf8)	12.00 ± 0.00 *^b^*	6.25	>100	>16	6.66 ± 0.57 *^a^*
**B_9_**	*Staphylococcus aureus* MDR (clinical strain, 136)	7.00 ± 0.00 *^a^*	12.5	100	>8	7.33 ± 0.57 *^a^*
**B_10_**	*Enterobacter cloacae* (clinical strain, 115)	14.33 ± 0.57 *^c^*	25	100	4	6.66 ± 0.57 *^a^*
**B_11_**	*Sphingomonas paucimobilis* (clinical strain, 144)	7.00 ± 0.00 *^a^*	12.5	>100	>8	7.66 ± 0.57 *^a^*
**B_12_**	*Acinetobacter baumannii* (clinical strain, 146)	19.33 ± 1.15 *^e^*	12.5	>100	>8	13.33 ± 0.57 *^b^*

* Inhibition zone around the disc with the methanolic extract (10 µL/disk; 3mg/disc) expressed as mean of three replicates (mm ± SD). Ampicillin (10 mg/mL). SD: standard deviation. ^a^: Minimal Inhibitory Concentration expressed as mg/mL. ^b^: Minimal Bactericidal Concentration expressed as mg/mL. The letters (*a–e*) indicate a significant difference between the inhibition zones of the sample according to the Duncan test (*p* < 0.05).

**Table 3 plants-09-01418-t003:** Growth inhibition zone, MIC and minimum fungicidal concentration (MFC) values obtained for fungal and yeast strains tested using disc diffusion and microdilution assays.

Code	Strain	*T. polium* Methanolic Extract				Amphotericin B
		Mean ± SD * (mm)	MIC ^a^	MFC ^b^	MFC/MIC Ratio	Mean ± SD (mm)
**Y_1_**	*Candida albicans* ATCC 10231	7.00 ± 0.00 *^b^*	12.5	50	2	22.66 ± 1.15 *^d^*
**Y_2_**	*Cryptococcus neoformans* ATCC 14116	7.00 ± 0.00 *^b^*	6.25	12.5	2	15.33 ± 0.57 *^c^*
**Y_3_**	*Candida vaginalis* (Clinical strain, 136)	10.33 ± 0.57 *^c^*	12.5	50	2	6.66 ± 0.57 *^a^*
**Y_4_**	*Candida albicans* (Clinical strain, 139)	7.00 ± 0.00 *^a^*	25	100	4	12.33 ± 0.57 *^b^*
**M_1_**	*Aspergillus fumigatus* ATCC 204305	6.00 ± 0.00 *^a^*	(-)	(-)	(-)	15.00 ± 1.00 *^c^*
**M_2_**	*Aspergillus niger*	6.00 ± 0.00 *^a^*	(-)	(-)	(-)	6.00 ± 0.00 *^a^*

* Inhibition zone around the disc impregnated with the methanolic extract (10 µL/disk) expressed as mean of three replicates (mm ± SD). SD: standard deviation. ^a^: Minimal Inhibitory Concentration expressed as mg/mL. ^b^: Minimal Fungicidal Concentration expressed as mg/mL. The letters (*a–e*) indicate a significant difference between the inhibition zones according to the Duncan test (*p* < 0.05).

**Table 4 plants-09-01418-t004:** Anti-swarming and anti-swimming activities of *T. polium* methanolic extract.

Sample	Swarming Inhibition (%)			Swimming Inhibition (%)		
	100	75	50	100	75	50
*T. polium* methanolic extract	23.66 ± 0.5	12.95 ± 1.5	-	35.25 ± 2.5	-	-
Concentration is expressed as µg/mL; (-): No activity; %: Percentage.						

**Table 5 plants-09-01418-t005:** Literature review of the main identified bioactive compounds in the *Teucrium* species and the solvent used.

*Teucrium* Species/Origin	Solvent Used	Main Identified Compounds	References
*T. polium* L. Serbia	Methanol	Phenolic acids (×10^−^^4^ µg/mL extract µg/mL extract):gallic acid (8.1 ± 0.2), vanillic acid (2.1 ± 0.3), caffeic acid (1.7 ± 0.2), chlorogenic acid (90.00 ± 0.2), and *p*-coumaric acid (30.0 ± 0.6). Flavonoids (×10^−^^4^ µg/mL extract µg/mL extract):catechin (235.0 ± 0.5), rutin (77.0 ± 0.1), myricetin (55.0 ± 0.1), luteolin (22.0 ± 0.6), quercetin (24.0 ± 0.4), and apigenin (157.0 ± 0.3).	[35]
*T. polium* L. Greece	Methanol/Water 62.5%	Phenolic compounds (mg/100g dry sample): tyrosol (0.42 ± 0.01), caffeic acid (0.65 ± 0.01), ferulic acid (0.95 ± 0.02), and luteolin (0.48 ± 0.01)	[36]
		o-Hydroxybenzoic acid, hydroxytyrosol, p-Hydroxybenzoic acid, vanillic acid, gentisic acid, ferulic acid, caffeic acid, 3-Nitro-phthalic acid, and quercetin.	
*T. polium* L. Egypt	Dichloromethane–methanol (1:1; *v*/*v*)	(1R,4S,10R) 10,11-dimethyl-dicyclohex-5 (6)-en-1,4-diol-7-one and (R)-mandelonitrile-b-laminaribioside.	[37]
*T. polium* L Italy	Methanol	Poliumoside B, poliumoside, teucardosid, 8-O-acetylharpagid, luteolin7-O-rutinoside, luteolin7-O-neohesperidoside, luteolin7-O-glucosied, luteolin 4′-O-glucoside, teulamifin, and teusalvin C.	[38]
*T. polium* L. R. Macedonia	Methanol	Hydroxycinnamic acid derivatives (2 compounds), phenylethanoid glycosides (12 compounds), flavonoid glycosides (11 compounds) and flavonoid aglycones (6 compounds).	[40]
*T. polium* L. Italy	Methanol	Poliumoside, apigenin, luteolin, montanin D, montanin E, teubutilin A, teuchamecrin C, teulamifin B, teupolin VI, teupolin VII, teupolin VIII, teupolin IX, teupolin X, teupolin XI, and teupolin XII.	[42]
*T. polium* L. Iran	Methanol	Rutin, apigenin, (3′, 6 dimethoxy apigenin, and 4′,7 dimethoxy apigenin.	[43]
*T. polium* L. Europe	Ethanol 70%	Rutin, apigenin, apigenin-4, 7-dimethylether, cirsimaritin, cirsiliol, luteolin, 6-hydroxyluteolin, luteolin-7-O-glucoside, salvigenin, apigenin 7-glucoside, eupatorin, apigenin-5 galloylglucoside, 3′,6- dimethoxy apigenin, 4′,7-dimethoxy apigenin, and salvigenin.	[44]
*T. polium* L. Egypt	Dichloromethane–methanol (1:1; *v*/*v*)	Sesquiterpenes: 4β,5α-epoxy-7αH-germacr-10(14)-en-6β-ol-1-one, 4β,5α-epoxy-7αH-germacr-10(14)-en,1β-hydroperoxyl,6β-ol, 4β,5β-epoxy-7αH-germacr-10(14)-en,1β-hydroperoxyl,6β-ol 4α,5β-epoxy-7αH-germacr-10(14)-en,1β-hydroperoxyl,6α-ol, 10a,1b;4b,5a-diepoxy-7aH-germacrm-6-ol, teucladiol, 4b,6b-dihydroxy-1a,5b(H)-guai-9-ene, oplopanone, oxyphyllenodiol A, eudesm-3-ene-1,6-diol, rel1b,3a,6b-trihydroxyeudesm-4-ene, arteincultone. Flavonoids: 7,40 -O-dimethylscutellar-ein(5,6-dihydroxy-7,40 -dimethoxyflavone) and salvigenin. Glycosides: teucardoside and poliumoside.	[90]
*T. polium* L. Saudi Arabia	Methanol	Larixol acetate, cepharanthine, Bis (2-hydroxypropyl) amine, 9-amino-nonanoic acid, 10-amino-decanoic acid, CMP-N-acetylneuraminic acid, selinidin, dihydrosamidin, triptonide, 10S,11R-Epoxy-punaglandin 4, rhoifolin, 3-hydroxy-3′,4′- dimethoxyflavone, sericetin diacetate, troxerutin, 1-dodecanoyl-sn-glycerol, harpagoside, koparin 2′-methyl ether, isotectorigenin, 7-methyl ether, 10-hydroxyloganin, 7-epiloganin tetraacetate, deoxyloganin tetraacetate, 8-epiiridodial glucoside tetraacetate, 16-hydroxy-4-carboxyretinoic acid, carapin-8 (9)-ene, khayanthone, 13R-hydroxy-9E,11Z octadecadienoic acid, b-D-glucopyranoside uronic acid, 6-(3-oxobutyl)-2-naphthalenyl, 16alpha,17beta-Estriol 3-(beta-D-glucuronide), and valtratum.	**This Study**

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
