# Peer review of "Phytochemical Screening, Antibacterial, Antifungal, Antiviral, Cytotoxic, and Anti-Quorum-Sensing Properties of Teucrium polium L. Aerial Parts Methanolic Extract"

_plants, 2020, doi:10.3390/plants9111418_

Round 1

Reviewer 1 Report

The novelty of the study is poor. It seem that Authors confirmed previously reported antibacterial and antiviral activity of T. polium. In lines 176-178 they stated: “In fact, our  results are in agreement with previous studies which demonstrated that methanolic extract exhibited  high antibacterial activity”.  

Phytochemical part of investigation is poorly described and documented. Lack of chromatographic details, e.g. type of column, mobile phase composition, flow rate, temperature. Lack of details on identification of compounds (mass spectra and fragmentation patterns). Figure 1 is completely unclear.

Section 2.1. – is not clear. On what basis the % different classes of compounds was assessed. Moreover, the values for different classes are the same – is it correct? A lot of various compounds are listed in lines 80-82 but on chromatogram they are not presented.

Discussion is very chaotic and should be more focused on the subject of the study. It need strong reedition. Sometimes the information are not associated with the subject of the study, e.g. lines 149-158 (the optimization of extraction in context of yield is not the aim of the work) etc..

There are also some minor errors, e. g. line 265: ratio 1/4? (should be 1:10), unnecessary or lack of capital letters etc

The quality of Fig.1 and 2 is very poor.

I strongly agree with Authors that “Further analysis are needed to correlate between the obtained results with the phytochemical composition of organic extract (…).”

Author Response

Point-by-point response to Reviewer 1:

 Dear Dr.  (Reviewer 1): Thank you for your consideration of our manuscript. We found the comments helpful, and believe our revised manuscript represents a significant improvement over our initial submission.

Comments and Suggestions for Authors

1-The novelty of the study is poor. It seem that Authors confirmed previously reported antibacterial and antiviral activity of T. polium. In lines 176-178 they stated: “In fact, our results are in agreement with previous studies which demonstrated that methanolic extract exhibited high antibacterial activity”.  

Dear colleague, Special thanks for this valuable comment. We modified this sentence and novelty of the present work is added in order to highlight the importance of our work and the main new findings added in the same topic using Teucrium polium plant species. In fact different techniques were used to characterize the active molecules from T. polium methanolic extract. We report in this paper the use of High resolution-Liquid chromatography coupled to mass spectrometry (HR-LCMS). Many compounds are described for the first time in T. polium methanolic extract. The antiviral activity was tested for the first time also against two viruses: CVB3 and HSV-2. Few studies reported the anti-quorum sensing activity of methanolic extract from Teucrium species.

For this purpose, the main objectives of the present investigation were to assess for the first time the phytochemical profile of T. polium methanolic extract from Hail region (Saudi Arabia) using High resolution-Liquid chromatography coupled to mass spectrometry (HR-LCMS). The antimicrobial activities of germander extract against a wide range and types of clinical bacterial and fungal isolates was also tested. The antiviral activity was tested for the first time against CVB3 and HSV-2. Inhibition of the production of some virulence factors in C. violaceum and P. aeruginosa PAO1 reporter strains was also investigated for the first time in the present work.

2-Phytochemical part of investigation is poorly described and documented. Lack of chromatographic details, e.g. type of column, mobile phase composition, flow rate, temperature. Lack of details on identification of compounds (mass spectra and fragmentation patterns). Figure 1 is completely unclear.

The HR-LCMS analysis of T. polium was analyzed by using UHPLC-PDA-Detector Mass spectrometer (HR-LCMS 1290 Infinity UHPLC System, 1260 Infinity Nano HPLC with Chipcube, 6550 iFunnel QTOFs), Agilent Technologies, USA. For chromatographic separation, an Agilent 1200 Series HPLC system (Agilent Technologies, USA) equipped with a binary gradient solvent pump, HiP Sampler, column oven and MS Q-TOF with Dual AJS ES Ion Source. Samples were separated on SB-C18 column (2.1 × 50 mm, 1.8-particle size; Agilent Technologies, USA) maintained at 25°C. The solvents used were: water containing 0.1% HCOOH and Acetonitrile containing 10% water and 0.1% HCOOH. The following gradient elution program at a flow rate of 0.3 mL min−1 was applied. MS detection was performed in MS Q-TOF Mass spectrometer (Agilent Technologies). Compounds were identified via their mass spectra and their unique mass fragmentation patterns. Compound Discoverer 2.1, ChemSpider and PubChem were used as the main tools for the identification of the phytochemical constituents.

Section 2.1. – is not clear. On what basis the % different classes of compounds was assessed. Moreover, the values for different classes are the same – is it correct? A lot of various compounds are listed in lines 80-82 but on chromatogram they are not presented.

All information regarding the phytochemical study of T. polium methanolic extract was added in details in Introduction and results and discussion sections.

Introduction part: Different techniques were used to identify the bioactive compounds in T. polium aerial parts extracted using different solvent (water, methanol, 65.5%- menthol-water, dichloromethane/methanol, acetone, ethanol 70%, and ethanol 95%) from different regions around the word [35-44]. A literature review of the phytochemical compounds isolated from T. polium aerial parts showed the identification of more than 130 molecules dominated by terpenoids (60%) [45]. The main identified compounds from T. polium roots, aerial parts and inflorescence are apigenin, luteolin, rutin, cirsiliol, cirsimaritin, salvigenin and eupatorin [45]. Five flavonoids were widely isolated from different Teucrium species from European countries namely: cirsiliol, cirsimaritin, cirsilineol, salvigenin and 5-hydroxy-6,7,3’,4’-tetramethoxyllavone [44]. Four new sesquiterpenes were identified in dichloromethane/methanol extract from T. polium, namely 4β,5α-epoxy-7αH-germacr-10(14)-en-6β-ol-1-one, 4β,5α-epoxy-7αH-germacr-10(14)-en,1β-hydroperoxyl,6β-ol, 4β,5β-epoxy-7αH-germacr-10(14)-en,1β-hydroperoxyl,6β-ol and 4α,5β-epoxy-7αH-germacr-10(14)-e [39].

Results parts: We added a table summarizing the list of compounds identified in

Our Results showed the identification of 29 molecules in T. polium methanolic extract by using HR-LCMS technique. The complete list of the identified compounds together with their respective chemical class are listed in table 1.

Table 2. Bioactive compounds and their chemical class identified in T. polium methanolic extract using HR-LCMS technique.

RT

Identified Compound Name

Chemical class

9.262

Larixol Acetate

Bicyclic labdane diterpenes

5.948

Cepharanthine

Alkaloid

1.062

Bis (2-hydroxypropyl) amine

Amino Alcohol

1.447

9-amino-nonanoic acid

Amino Fatty Acid

3.807

10-Amino-decanoic acid

Amino Fatty Acid

6.329

CMP-N-acetylneuraminic acid

Amino Sugar

8.848

Selinidin

Coumarin Derivative

10.779

Dihydrosamidin

Coumarins

9.807

Triptonide

Diterpene triepoxide

11.022

10S,11R-Epoxy-punaglandin 4

Eicosanoid

6.376

Rhoifolin

Flavonoid

13.274

3-hydroxy-3',4'- Dimethoxyflavone

Flavonoid

7.838

Sericetin diacetate

Flavonol

5.96

Troxerutin

Flavonol

20.37

1-dodecanoyl-sn-glycerol

Glycerolipid

9.1

Harpagoside

Iridoid Glycoside

10.036

Koparin 2'-Methyl Ether

Isoflavonoid

12.149

Isotectorigenin, 7- Methyl ether

Isoflavonoid

0.945

10-Hydroxyloganin

Isoprenoid

4.739

7-Epiloganin tetraacetate

Isoprenoid

6.319

Deoxyloganin tetraacetate

Isoprenoid

9.126

8-Epiiridodial glucoside tetraacetate

Isoprenoid

11.28

16-Hydroxy-4-carboxyretinoic Acid

Isoprenoid

8.198

Carapin-8 (9)-Ene

Limonoid

18.427

Khayanthone

Limonoid

1.046

13R-hydroxy-9E,11Z octadecadienoic acid

Octadecanoid

5.342

b-D-Glucopyranoside uronic acid, 6-(3-oxobutyl)-2- naphthalenyl

Organic Acid, Phenol

11.279

16alpha,17beta-Estriol 3-(beta-D-glucuronide)

Steroidal glycosides

9.525

Valtratum

Terpene

In fat, T. polium methanolic extract was dominated by terpenes and their derivatives, fatty acids and their derivatives, polyphenols, flavonoids and derivatives, coumarines, amino acid derivatives, and alkaloids. The tested extract was dominated by 3R-hydroxy-9E,11Z-octadecadienoic acid,  valtratum, dihydrosamidin, and cepharanthine. The chromatograms (figure 2) showed respectively major peaks indicating the presence of various phytochemical constituents.

            Discussion section: A table summarizing different molecules isolated from Teucrium extracts are reported, especially from methanolic one.

Table 4 summarize some important bioactive molecules isolated from Teucrium species, mainly T. polium using various identification techniques and different extraction procedures [35-44].

Table 4. Literature review of the main identified bioactive compounds in Teucrium species and the solvent used.

Teucrium species

/Origin

Solvent Used

Main identified compounds

References

T. polium L.

 Serbia

Methanol

Phenolic acids (×10–4 µg/mL extract µg/mL extract): gallic acid (8.1±0.2), vanillic acid (2.1±0.3), caffeic acid (1.7 ± 0.2), chlorogenic acid (90.00±0.2), and p-coumaric acid (30.0±0.6).

Flavonoids (×10–4 µg/mL extract µg/mL extract): catechin (235.0±0.5), rutin (77.0±0.1), myricetin (55.0±0.1), luteolin (22.0±0.6), quercetin (24.0±0.4), and apigenin (157.0 ± 0.3).

[35]

T. scordium L.

Serbia

Phenolic acids (×10–4 µg/mL extract µg/mL extract): vanillic acid (501.1 ± 0.6), p-coumaric acid (111.5 ± 0.1), and sinapinic acid (26.9 ± 0.2).

Flavonoids (×10–4 µg/mL extract µg/mL extract): catechin (32.5 ± 0.6) and rutin (24.7±0.1).

T. polium L.

Greece

Methanol/Water

62.5%

Phenolic compounds (mg/100g dry sample):  tyrosol (0.42 ± 0.01),  caffeic acid (0.65 ± 0.01),  ferulic acid (0.95 ± 0.02), and  luteolin (0.48 ± 0.01)

[36]

o-Hydroxybenzoic acid,  hydroxytyrosol,  p-Hydroxybenzoic acid,  vanillic acid,  gentisic acid,  ferulic acid,  caffeic acid,  3-Nitro-phthalic acid , and  quercetin.

T. polium L.

Egypt

Dichloromethane–methanol

 (1:1; v/v)

(1R,4S,10R) 10,11-dimethyl-dicyclohex-5 (6)-en-1,4-diol-7-one and (R)-mandelonitrile-b-laminaribioside.

[37]

T. polium L

Italy

Methanol

Poliumoside B, poliumoside, teucardosid, 8-O-acetylharpagid, luteolin7-O-rutinoside,  luteolin7-O-neohesperidoside,  luteolin7-O-glucosied, luteolin 4´-O-glucoside, teulamifin, and teusalvin C.

[38]

T. stocksianum Bioss.

Italy

Water

Flavonoids, Saponins, Reducing sugars, Terpenoids, and Tannins.

[39]

T. polium L.

R. Macedonia

Methanol

Hydroxycinnamic acid derivatives (2 compounds), phenylethanoid glycosides (12 compounds), flavonoid glycosides (11 compounds) and flavonoid aglycones (6 compounds).

[40]

T. marum

Italy

Acetone or ethanol 95%

Teumarin 1 and teumarin B 2.

[41]

T. polium L.

Italy

Methanol

Poliumoside, apigenin, luteolin, montanin D, montanin E, teubutilin A, teuchamecrin C, teulamifin B, teupolin VI,   teupolin VII,  teupolin VIII, teupolin IX, teupolin X, teupolin XI, and teupolin XII.

[42]

T. polium L.

Iran

Methanol

Rutin, apigenin, (3´, 6 dimethoxy apigenin, and 4´ ,7 dimethoxy apigenin.

[43]

T. polium L.

Europe

Ethanol

70%

Rutin, apigenin, apigenin-4, 7-dimethylether, cirsimaritin, cirsiliol, luteolin, 6-hydroxyluteolin, luteolin-7-O-glucoside, salvigenin, apigenin 7-glucoside, eupatorin, apigenin-5 galloylglucoside, 3′,6- dimethoxy apigenin, 4′,7- dimethoxy apigenin and salvigenin.

[44]

T. polium L.

Egypt

Dichloromethane–methanol

 (1:1; v/v)

Sesquiterpenes: 4β,5α-epoxy-7αH-germacr-10(14)-en-6β-ol-1-one, 4β,5α-epoxy-7αH-germacr-10(14)-en,1β-hydroperoxyl,6β-ol, 4β,5β-epoxy-7αH-germacr-10(14)-en,1β-hydroperoxyl,6β-ol 4α,5β-epoxy-7αH-germacr-10(14)-en,1β-hydroperoxyl,6α-ol, 10a,1b;4b,5a-diepoxy-7aH-germacrm-6-ol, teucladiol, 4b,6b-dihydroxy-1a,5b(H)-guai-9-ene, oplopanone, oxyphyllenodiol A, eudesm-3-ene-1,6-diol, rel1b,3a,6b-trihydroxyeudesm-4-ene, arteincultone. Flavonoids: 7,40 -O-dimethylscutellar-ein(5,6-dihydroxy-7,40 -dimethoxyflavone) and salvigenin. Glycosides: teucardoside and poliumoside.

[90]

T. polium L.

Saudi Arabia

Methanol

13R-hydroxy-9E,11Z-octadecadienoic acid, rhoifolin, sericetin diacetate, selinidin, harpagoside, valtratum, triptonide, koparin 2'-methyl ether, dihydrosamidin, 10S,11R-epoxy-punaglandin, khayanthone, 4, 16alpha, 17beta-Estriol 3-(beta-D-glucuronide), 10-hydroxyloganin, 7-epiloganin tetraacetate, cepharanthine, deoxyloganin tetraacetate, carapin-8 (9)-Ene, and 1-dodecanoyl-sn-glycerol.

This Study

In the present work, we reported the identification of a new undescribed compounds in T. polium aerial parts methanolic extract using HR-LCMS technique: 13R-hydroxy-9E,11Z-octadecadienoic acid, rhoifolin, sericetin diacetate, selinidin, harpagoside, valtratum, triptonide, koparin 2'-methyl ether, dihydrosamidin, 10S,11R-epoxy-punaglandin, 4, 16alpha, 17beta-Estriol 3-(beta-D-glucuronide), khayanthone, 10-hydroxyloganin, 7-epiloganin tetraacetate, cepharanthine, deoxyloganin tetraacetate, carapin-8 (9)-Ene, and 1-dodecanoyl-sn-glycerol. In previous work, Pacifico et al. [42] reported the identification of poliumoside, apigenin, luteolin, montanin D, montanin E, teubutilin A, teuchamecrin C, teulamifin B, teupolin VI,   teupolin VII, teupolin VIII, teupolin IX, teupolin X, teupolin XI, and teupolin XII from T. polium methanolic extract from Italy. Using the same solvent, De Marino et al. [40] identified poliumoside B, poliumoside, teucardosid, 8-O-acetylharpagid, luteolin7-O-rutinoside, luteolin 7-O-neohesperidoside, luteolin 7-O-glucosied, luteolin 4´-O-glucoside, teulamifin, and teusalvin C as main bioactive compounds from T. polium aerial parts from Italy. While, Sharififar et al. [43] identified 3´,6 dimethoxy apigenin, rutin, apigenin, and 4´,7 dimethoxy apigenin as the main bioactive molecules in T. polium methanolic extract from Iran.

Discussion is very chaotic and should be more focused on the subject of the study. It need strong reedition. Sometimes the information are not associated with the subject of the study, e.g. lines 149-158 (the optimization of extraction in context of yield is not the aim of the work) etc..

Dear Dr., special thanks for the comment: In fact, we discussed the yield of extraction because we have used only methanol as solvent for the extraction procedure. We previously reported that this solvent possess the highest yield of extraction and the best biological activities.  Many works described the use of ethanol as solvent for the bioactive compounds isolation from Teucrium species. So, we decided to compare our findings in comparison with previous reports as summarized in Table 4.

Here are also some minor errors, e. g. line 265: ratio 1/4? (should be 1:10), unnecessary or lack of capital letters etc

The whole manuscript was deeply revised in order to avoid spelling and grammatical errors.

The quality of Fig.1 and 2 is very poor.

The figure 1 was deleted and figure 2 was updated.

I strongly agree with Authors that “Further analysis are needed to correlate between the obtained results with the phytochemical composition of organic extract (…).”

Reviewer 2 Report

Manuscript title: Phytochemical screening, antibacterial, antifungal, antiviral, cytotoxic and anti-quorum sensing properties of Teucrium polium L. aerial parts methanolic extract.

Authors: Mousa Alreshidi , Emira Noumi , Lamjed Bouslama , Ozgur Ceylan , Vajid N. Veettil , Mohd Adnan , Corina Danciu , Salem Elkahoui , Riadh Badraoui , Khalid A. Al-Motair , Mitesh Patel , Vincenzo De Feo , Mejdi Snoussi 

The manuscript is fairly well written and includes a great deal of information, which is reflected in the significant number of references listed.

Abbreviations should be defined the first time they appear in the abstract.

The full name of the plant should be mention in the abstract.

All the Latin names of the plants, viruses, bacteria, and fungi should be written in italic. Check the whole manuscript!

The keywords should be specific to the article but not the same as in the title.

23 position for the first paragraph of the introduction part? Isn't that an unnecessary multiplication of reference positions?

More details on the bioactive compounds found in the studied plants should be mentioned in the introduction part.

The phytochemical part of the result section should be more described.

Figure 3 The description of the Figure 3 should contain the name of the viability test used.

Table 1 The description of the Table 1 is confusing… the extract or essential oil was used?

Table 2 The description of the Table 2 is confusing… the extract or essential oil was used? Bactericidal or fungicidal? Against bacteria or against fungi?

The geographical position of the plant habitat as well as the voucher specimen should be added to the experimental/ methodology part.

What was the positive and negative control in the cytotoxic activity experiment?

References should be described as recommended by the journal.

Moderate English changes are required.

Author Response

Dear Dr.  (Reviewer 2): Thank you for your consideration of our manuscript. We found the comments helpful, and believe our revised manuscript represents a significant improvement over our initial submission.

Comments and Suggestions for Authors

Manuscript title: Phytochemical screening, antibacterial, antifungal, antiviral, cytotoxic and anti-quorum sensing properties of Teucrium polium L. aerial parts methanolic extract.

Authors: Mousa Alreshidi , Emira Noumi , Lamjed Bouslama , Ozgur Ceylan , Vajid N. Veettil , Mohd Adnan , Corina Danciu , Salem Elkahoui , Riadh Badraoui , Khalid A. Al-Motair , Mitesh Patel , Vincenzo De Feo , Mejdi Snoussi 

The manuscript is fairly well written and includes a great deal of information, which is reflected in the significant number of references listed.

  1. Abbreviations should be defined the first time they appear in the abstract.

All abbreviations were checked and were defined the first time they appear in the text.

  1. The full name of the plant should be mention in the abstract.

The full name of the studied plant was added in the abstract: Teucrium polium L.

  1. All the Latin names of the plants, viruses, bacteria, and fungi should be written in italic. Check the whole manuscript!

All Latin names for viruses, bacteria, and plants were checked and rewritten in italic in whole manuscript.

  1. The keywords should be specific to the article but not the same as in the title.

Keywords were modified as suggested. Teucrium polium L., aerial parts; bioactive compounds; HR-LCMS; biological activities.

  1. 23 position for the first paragraph of the introduction part? Isn't that an unnecessary multiplication of reference positions?

Dear Colleague, special thanks for your comment.

In fact, we used new references to describe the interest of natural compounds from plants in different countries around the word.

 For this reason, we cited six references in the first three lines.

Aromatic and medicinal plants history is associated with the evolution of civilizations and occupied a crucial place in medicine, cosmetology and culinary preparations in China, India, the Middle East, Egypt, Saudi Arabia and Greece [1-6].

  1. More details on the bioactive compounds found in the studied plants should be mentioned in the introduction part.

More details were added in the new version of the manuscript. All these modifications are highlighted in yellow in the text.

Different techniques were used to identify the bioactive compounds in T. polium aerial parts extracted using different solvent (water, methanol, 65.5%- menthol-water, dichloromethane/methanol, acetone, ethanol 70%, and ethanol 95%) from different regions around the word [35-44]. A literature review of the phytochemical compounds isolated from T. polium aerial parts showed the identification of more than 130 molecules dominated by terpenoids (60%) [45]. The main identified compounds from T. polium roots, aerial parts and inflorescence are apigenin, luteolin, rutin, cirsiliol, cirsimaritin, salvigenin and eupatorin [45]. Five flavonoids were widely isolated from different Teucrium species from European countries namely: cirsiliol, cirsimaritin, cirsilineol, salvigenin and 5-hydroxy-6,7,3’,4’-tetramethoxyllavone [44]. Four new sesquiterpenes were identified in dichloromethane/methanol extract from T. polium, namely 4β,5α-epoxy-7αH-germacr-10(14)-en-6β-ol-1-one, 4β,5α-epoxy-7αH-germacr-10(14)-en,1β-hydroperoxyl,6β-ol, 4β,5β-epoxy-7αH-germacr-10(14)-en,1β-hydroperoxyl,6β-ol and 4α,5β-epoxy-7αH-germacr-10(14)-e [39].

  1. The phytochemical part of the result section should be more described.

More details were added in the new version of the manuscript. All these modifications are highlighted in yellow in the text.

In a systematic review, Harborne et al. [44] reported the distribution of free flavone aglycones and flaoonoid glycosides from 42 European taxa of the genus Teucrium (75% ethanol extract from aerial parts). These authors reported the identification of 5,7-Dihydroxytkvones, 6-hydroxyBavones; 6-methoxytIavoncs; flavonols; 8-hydroxyflavones; cirsiliol; cirsimaritin; citsilineol; 5-hydroxy-6,7,3’,4’-tetramethoxyllavone; salvigenin; luteolin; apigenin; diosmetiq, hydroxyluteolin; cirsimariti, quercetin; isorhamnetin; hypolaetin; isoscutellarein, vicenin-2; 5,7dihydroxytlavone glycosides; 6-hydroxytIavone glycosides; I-hydroxyflavone glycoside, 6-methoxyflavone glycosides; tlavonol glycoside, apigenin-7-glucoside, apigenin 7-rutinoside; luteolin 7-glucoside; lutcolin 7-rutinoside; luteolin 7-sambubiosidc; diosmetin 7-rutinosidc; 6-OH-luteolin 7-glucoside; 6-OH-luteolin 7-rhamnoside; isoscutellarein 7-allosylglucoside, hypolaetin 7-allosylglucoside; cirsimaritin 4´-glucoside; quercetin 3-glucoside; quercetin 3-rutinoside; isorhanmetin 3-glucoside; and isorhamnetin 3-rutinoside.

Table 4 summarize some important bioactive molecules isolated from Teucrium species, mainly T. polium using various identification techniques and different extraction procedures [35-44].

Table 4. Literature review of the main identified bioactive compounds in Teucrium species and the solvent used.

Teucrium species

/Origin

Solvent Used

Main identified compounds

References

T. polium L.

 Serbia

Methanol

Phenolic acids (×10–4 µg/mL extract µg/mL extract): gallic acid (8.1±0.2), vanillic acid (2.1±0.3), caffeic acid (1.7 ± 0.2), chlorogenic acid (90.00±0.2), and p-coumaric acid (30.0±0.6).

Flavonoids (×10–4 µg/mL extract µg/mL extract): catechin (235.0±0.5), rutin (77.0±0.1), myricetin (55.0±0.1), luteolin (22.0±0.6), quercetin (24.0±0.4), and apigenin (157.0 ± 0.3).

[35]

T. scordium L.

Serbia

Phenolic acids (×10–4 µg/mL extract µg/mL extract): vanillic acid (501.1 ± 0.6), p-coumaric acid (111.5 ± 0.1), and sinapinic acid (26.9 ± 0.2).

Flavonoids (×10–4 µg/mL extract µg/mL extract): catechin (32.5 ± 0.6) and rutin (24.7±0.1).

T. polium L.

Greece

Methanol/Water

62.5%

Phenolic compounds (mg/100g dry sample):  tyrosol (0.42 ± 0.01),  caffeic acid (0.65 ± 0.01),  ferulic acid (0.95 ± 0.02), and  luteolin (0.48 ± 0.01)

[36]

o-Hydroxybenzoic acid,  hydroxytyrosol,  p-Hydroxybenzoic acid,  vanillic acid,  gentisic acid,  ferulic acid,  caffeic acid,  3-Nitro-phthalic acid , and  quercetin.

T. polium L.

Egypt

Dichloromethane–methanol

 (1:1; v/v)

(1R,4S,10R) 10,11-dimethyl-dicyclohex-5 (6)-en-1,4-diol-7-one and (R)-mandelonitrile-b-laminaribioside.

[37]

T. polium L

Italy

Methanol

Poliumoside B, poliumoside, teucardosid, 8-O-acetylharpagid, luteolin7-O-rutinoside,  luteolin7-O-neohesperidoside,  luteolin7-O-glucosied, luteolin 4´-O-glucoside, teulamifin, and teusalvin C.

[38]

T. stocksianum Bioss.

Italy

Water

Flavonoids, Saponins, Reducing sugars, Terpenoids, and Tannins.

[39]

T. polium L.

R. Macedonia

Methanol

Hydroxycinnamic acid derivatives (2 compounds), phenylethanoid glycosides (12 compounds), flavonoid glycosides (11 compounds) and flavonoid aglycones (6 compounds).

[40]

T. marum

Italy

Acetone or ethanol 95%

Teumarin 1 and teumarin B 2.

[41]

T. polium L.

Italy

Methanol

Poliumoside, apigenin, luteolin, montanin D, montanin E, teubutilin A, teuchamecrin C, teulamifin B, teupolin VI,   teupolin VII,  teupolin VIII, teupolin IX, teupolin X, teupolin XI, and teupolin XII.

[42]

T. polium L.

Iran

Methanol

Rutin, apigenin, (3´, 6 dimethoxy apigenin, and 4´ ,7 dimethoxy apigenin.

[43]

T. polium L.

Europe

Ethanol

70%

Rutin, apigenin, apigenin-4, 7-dimethylether, cirsimaritin, cirsiliol, luteolin, 6-hydroxyluteolin, luteolin-7-O-glucoside, salvigenin, apigenin 7-glucoside, eupatorin, apigenin-5 galloylglucoside, 3′,6- dimethoxy apigenin, 4′,7- dimethoxy apigenin and salvigenin.

[44]

T. polium L.

Egypt

Dichloromethane–methanol

 (1:1; v/v)

Sesquiterpenes: 4β,5α-epoxy-7αH-germacr-10(14)-en-6β-ol-1-one, 4β,5α-epoxy-7αH-germacr-10(14)-en,1β-hydroperoxyl,6β-ol, 4β,5β-epoxy-7αH-germacr-10(14)-en,1β-hydroperoxyl,6β-ol 4α,5β-epoxy-7αH-germacr-10(14)-en,1β-hydroperoxyl,6α-ol, 10a,1b;4b,5a-diepoxy-7aH-germacrm-6-ol, teucladiol, 4b,6b-dihydroxy-1a,5b(H)-guai-9-ene, oplopanone, oxyphyllenodiol A, eudesm-3-ene-1,6-diol, rel1b,3a,6b-trihydroxyeudesm-4-ene, arteincultone. Flavonoids: 7,40 -O-dimethylscutellar-ein(5,6-dihydroxy-7,40 -dimethoxyflavone) and salvigenin. Glycosides: teucardoside and poliumoside.

[90]

T. polium L.

Saudi Arabia

Methanol

13R-hydroxy-9E,11Z-octadecadienoic acid, rhoifolin, sericetin diacetate, selinidin, harpagoside, valtratum, triptonide, koparin 2'-methyl ether, dihydrosamidin, 10S,11R-epoxy-punaglandin, khayanthone, 4, 16alpha, 17beta-Estriol 3-(beta-D-glucuronide), 10-hydroxyloganin, 7-epiloganin tetraacetate, cepharanthine, deoxyloganin tetraacetate, carapin-8 (9)-Ene, and 1-dodecanoyl-sn-glycerol.

This Study

In the present work, we reported the identification of a new undescribed compounds in T. polium aerial parts methanolic extract using HR-LCMS technique: 13R-hydroxy-9E,11Z-octadecadienoic acid, rhoifolin, sericetin diacetate, selinidin, harpagoside, valtratum, triptonide, koparin 2'-methyl ether, dihydrosamidin, 10S,11R-epoxy-punaglandin, 4, 16alpha, 17beta-Estriol 3-(beta-D-glucuronide), khayanthone, 10-hydroxyloganin, 7-epiloganin tetraacetate, cepharanthine, deoxyloganin tetraacetate, carapin-8 (9)-Ene, and 1-dodecanoyl-sn-glycerol. In previous work, Pacifico et al. [42] reported the identification of poliumoside, apigenin, luteolin, montanin D, montanin E, teubutilin A, teuchamecrin C, teulamifin B, teupolin VI,   teupolin VII, teupolin VIII, teupolin IX, teupolin X, teupolin XI, and teupolin XII from T. polium methanolic extract from Italy. Using the same solvent, De Marino et al. [40] identified poliumoside B, poliumoside, teucardosid, 8-O-acetylharpagid, luteolin7-O-rutinoside, luteolin 7-O-neohesperidoside, luteolin 7-O-glucosied, luteolin 4´-O-glucoside, teulamifin, and teusalvin C as main bioactive compounds from T. polium aerial parts from Italy. While, Sharififar et al. [43] identified 3´,6 dimethoxy apigenin, rutin, apigenin, and 4´,7 dimethoxy apigenin as the main bioactive molecules in T. polium methanolic extract from Iran.

  1. Figure 3 The description of the Figure 3 should contain the name of the viability test used.

The name of the viability test used was added.

  1. Table 1 The description of the Table 1 is confusing… the extract or essential oil was used?

*Inhibition zone around the discs impregnated with the methanolic extract (10 µl/disk) expressed as mean of three replicates (mm ± SD). SD: standard deviation. a: Minimal Inhibitory Concentration. b: Minimal Bactericidal Concentration. The letters (a–e) indicate a significant difference between the inhibition zones of the sample according to the Duncan test (p < 0.05).

  1. Table 2 The description of the Table 2 is confusing… the extract or essential oil was used? Bactericidal or fungicidal? Against bacteria or against fungi?

*Inhibition zone around the discs impregnated with the methanolic extract (10 µl/disk) expressed as mean of three replicates (mm ± SD). SD: standard deviation. a: Minimal Inhibitory Concentration. b: Minimal Fungicidal Concentration. The letters (a–e) indicate a significant difference between the inhibition zones according to the Duncan test (p < 0.05).

  1. The geographical position of the plant habitat as well as the voucher specimen should be added to the experimental/ methodology part.

The plant material was collected in October 2019 from a nursery belonging to the Ministry of Agriculture in Hail region, Saudi Arabia. A voucher specimen (AN 01) was deposited in the Department of Biology (College of Science, University of Hail, KSA).

  1. What was the positive and negative control in the cytotoxic activity experiment?

Cell culture without extract was used as negative control. The positive control is not required for the cytotoxicity assay.

The cytotoxicity of the T. polium extract was evaluated on VERO (African green monkey kidney) cells line. 104 Vero cells per well were seeded in 96-well plates and grown in Eagle’s minimum essential medium (MEM) supplemented with 5% fetal bovine serum (FBS), 100 units/ml of penicillin, 100 g/mL of streptomycin, 0.25 g/mL of Amphotericin B at 37â—¦C under 5% CO2 atmosphere. After 24 h incubation, confluent monolayers of Vero cells were exposed to decreasing concentration of the extract (from 3333 µg/mL to 1 g/mL) diluted in MEM supplemented with 2% FBS. After 48 h incubation, cell viability was observed in an inverted microscope. The 50% cytotoxic concentration (CC50), defined as the concentration of the extract able to reduce of 50% the cell viability, was determined by regression analysis in comparison to negative control.

  1. References should be described as recommended by the journal.

All references were checked and updated according to the journal guidelines.

  1. Moderate English changes are required.

The whole manuscript was revised in order to avoid spelling and grammatical errors.

Reviewer 3 Report

The manuscript described the antimicrobial, antiviral and citotoxic activity of Teucrium polium L. aerial parts. Many studies have already studied this effect so the authors should underline the novelty of this work, especially in the discussion section, In fact the authors mentioned several previous studies but the comparison withobtained results should be better explained in the relative section. Moreover, the choice of extraction solvent is not clear: previous studies reported a higher yield with ethanol, please justify the use of methanol. On the other hand, about the extraction procedure, time and temperature should be indicated as well as in other experiments like cytoxic and antibacterial activity. The method used to compound identification is missing. The abbreviations should be explain when mentioned the first time in the paper. The use of VERO cell lines should be motivated. Results of cytotoxic effect should be better discussed and possibly correlated with other activities.

Author Response

Dear Dr.  (Reviewer 3): Thank you for your consideration of our manuscript. We found the comments helpful, and believe our revised manuscript represents a significant improvement over our initial submission.

The manuscript described the antimicrobial, antiviral and cytotoxic activity of Teucrium polium L. aerial parts.

  • Many studies have already studied this effect so the authors should underline the novelty of this work, especially in the discussion section, In fact the authors mentioned several previous studies but the comparison with obtained results should be better explained in the relative section.

More details were added in the new version of the manuscript in the Itroduction part. All these modifications are highlighted in yellow in the text.

Different techniques were used to identify the bioactive compounds in T. polium aerial parts extracted using different solvent (water, methanol, 65.5%- menthol-water, dichloromethane/methanol, acetone, ethanol 70%, and ethanol 95%) from different regions around the word [35-44]. A literature review of the phytochemical compounds isolated from T. polium aerial parts showed the identification of more than 130 molecules dominated by terpenoids (60%) [45]. The main identified compounds from T. polium roots, aerial parts and inflorescence are apigenin, luteolin, rutin, cirsiliol, cirsimaritin, salvigenin and eupatorin [45]. Five flavonoids were widely isolated from different Teucrium species from European countries namely: cirsiliol, cirsimaritin, cirsilineol, salvigenin and 5-hydroxy-6,7,3’,4’-tetramethoxyllavone [44]. Four new sesquiterpenes were identified in dichloromethane/methanol extract from T. polium, namely 4β,5α-epoxy-7αH-germacr-10(14)-en-6β-ol-1-one, 4β,5α-epoxy-7αH-germacr-10(14)-en,1β-hydroperoxyl,6β-ol, 4β,5β-epoxy-7αH-germacr-10(14)-en,1β-hydroperoxyl,6β-ol and 4α,5β-epoxy-7αH-germacr-10(14)-e [39].

More details were added in the new version of the manuscript in the Discussion part. All these modifications are highlighted in yellow in the text.

In a systematic review, Harborne et al. [44] reported the distribution of free flavone aglycones and flaoonoid glycosides from 42 European taxa of the genus Teucrium (75% ethanol extract from aerial parts). These authors reported the identification of 5,7-Dihydroxytkvones, 6-hydroxyBavones; 6-methoxytIavoncs; flavonols; 8-hydroxyflavones; cirsiliol; cirsimaritin; citsilineol; 5-hydroxy-6,7,3’,4’-tetramethoxyllavone; salvigenin; luteolin; apigenin; diosmetiq, hydroxyluteolin; cirsimariti, quercetin; isorhamnetin; hypolaetin; isoscutellarein, vicenin-2; 5,7dihydroxytlavone glycosides; 6-hydroxytIavone glycosides; I-hydroxyflavone glycoside, 6-methoxyflavone glycosides; tlavonol glycoside, apigenin-7-glucoside, apigenin 7-rutinoside; luteolin 7-glucoside; lutcolin 7-rutinoside; luteolin 7-sambubiosidc; diosmetin 7-rutinosidc; 6-OH-luteolin 7-glucoside; 6-OH-luteolin 7-rhamnoside; isoscutellarein 7-allosylglucoside, hypolaetin 7-allosylglucoside; cirsimaritin 4´-glucoside; quercetin 3-glucoside; quercetin 3-rutinoside; isorhanmetin 3-glucoside; and isorhamnetin 3-rutinoside.

In the present work, we reported the identification of a new undescribed compounds in T. polium aerial parts methanolic extract using HR-LCMS technique: 13R-hydroxy-9E,11Z-octadecadienoic acid, rhoifolin, sericetin diacetate, selinidin, harpagoside, valtratum, triptonide, koparin 2'-methyl ether, dihydrosamidin, 10S,11R-epoxy-punaglandin, 4, 16alpha, 17beta-Estriol 3-(beta-D-glucuronide), khayanthone, 10-hydroxyloganin, 7-epiloganin tetraacetate, cepharanthine, deoxyloganin tetraacetate, carapin-8 (9)-Ene, and 1-dodecanoyl-sn-glycerol. In previous work, Pacifico et al. [42] reported the identification of poliumoside, apigenin, luteolin, montanin D, montanin E, teubutilin A, teuchamecrin C, teulamifin B, teupolin VI,   teupolin VII, teupolin VIII, teupolin IX, teupolin X, teupolin XI, and teupolin XII from T. polium methanolic extract from Italy. Using the same solvent, De Marino et al. [40] identified poliumoside B, poliumoside, teucardosid, 8-O-acetylharpagid, luteolin7-O-rutinoside, luteolin 7-O-neohesperidoside, luteolin 7-O-glucosied, luteolin 4´-O-glucoside, teulamifin, and teusalvin C as main bioactive compounds from T. polium aerial parts from Italy. While, Sharififar et al. [43] identified 3´,6 dimethoxy apigenin, rutin, apigenin, and 4´,7 dimethoxy apigenin as the main bioactive molecules in T. polium methanolic extract from Iran.

  • Moreover, the choice of extraction solvent is not clear: previous studies reported a higher yield with ethanol, please justify the use of methanol.

Dear Colleague, Yes I agree that many reports have tested ethanol as solvent (extraction). But there is also many reports describing the use of methanol for the extraction of bioactive molecules from Teucrium polium aerial parts.

Table 4 summarize some important bioactive molecules isolated from Teucrium species, mainly T. polium using various identification techniques and different extraction procedures [35-44].

Table 4. Literature review of the main identified bioactive compounds in Teucrium species and the solvent used.

Teucrium species/

Origin

Solvent Used

Main identified compounds

Reference

T. polium L.

 Serbia

Methanol

Phenolic acids (×10–4 µg/mL extract µg/mL extract): gallic acid (8.1±0.2), vanillic acid (2.1±0.3), caffeic acid (1.7 ± 0.2), chlorogenic acid (90.00±0.2), and p-coumaric acid (30.0±0.6).

Flavonoids (×10–4 µg/mL extract µg/mL extract): catechin (235.0±0.5), rutin (77.0±0.1), myricetin (55.0±0.1), luteolin (22.0±0.6), quercetin (24.0±0.4), and apigenin (157.0 ± 0.3).

35

T. scordium L.

Serbia

Methanol

Phenolic acids (×10–4 µg/mL extract µg/mL extract): vanillic acid (501.1 ± 0.6), p-coumaric acid (111.5 ± 0.1), and sinapinic acid (26.9 ± 0.2).

Flavonoids (×10–4 µg/mL extract µg/mL extract): catechin (32.5 ± 0.6) and rutin (24.7±0.1).

T. polium L.

Greece

Methanol/Water

62.5%

Phenolic compounds (mg/100g dry sample):  tyrosol (0.42 ± 0.01),  caffeic acid (0.65 ± 0.01),  ferulic acid (0.95 ± 0.02), and  luteolin (0.48 ± 0.01)

36

o-Hydroxybenzoic acid,  hydroxytyrosol,  p-Hydroxybenzoic acid,  vanillic acid,  gentisic acid,  ferulic acid,  caffeic acid,  3-Nitro-phthalic acid , and  quercetin.

T. polium L.

Egypt

Dichloromethane–methanol

 (1:1; v/v)

(1R,4S,10R) 10,11-dimethyl-dicyclohex-5 (6)-en-1,4-diol-7-one and (R)-mandelonitrile-b-laminaribioside.

37

T. polium L

Italy

Methanol

Poliumoside B, poliumoside, teucardosid, 8-O-acetylharpagid, luteolin7-O-rutinoside,  luteolin7-O-neohesperidoside,  luteolin7-O-glucosied, luteolin 4´-O-glucoside, teulamifin, and teusalvin C.

38

T. stocksianum Bioss.

Italy

Water

Flavonoids, Saponins, Reducing sugars, Terpenoids, and Tannins.

39

T. polium L.

R. Macedonia

Methanol

Hydroxycinnamic acid derivatives (2 compounds), phenylethanoid glycosides (12 compounds), flavonoid glycosides (11 compounds) and flavonoid aglycones (6 compounds).

40

T. marum

Italy

Acetone or ethanol 95%

Teumarin 1 and teumarin B 2.

41

T. polium L.

Italy

Methanol

Poliumoside, apigenin, luteolin, montanin D, montanin E, teubutilin A, teuchamecrin C, teulamifin B, teupolin VI,   teupolin VII,  teupolin VIII, teupolin IX, teupolin X, teupolin XI, and teupolin XII.

42

T. polium L.

Iran

Methanol

Rutin, apigenin, (3´, 6 dimethoxy apigenin ), and 4´ ,7 dimethoxy apigenin.

43

T. polium L.

Egypt

Dichloromethane–methanol

 (1:1; v/v)

Sesquiterpenes: 4β,5α-epoxy-7αH-germacr-10(14)-en-6β-ol-1-one, 4β,5α-epoxy-7αH-germacr-10(14)-en,1β-hydroperoxyl,6β-ol, 4β,5β-epoxy-7αH-germacr-10(14)-en,1β-hydroperoxyl,6β-ol 4α,5β-epoxy-7αH-germacr-10(14)-en,1β-hydroperoxyl,6α-ol, 10a,1b;4b,5a-diepoxy-7aH-germacrm-6-ol, teucladiol, 4b,6b-dihydroxy-1a,5b(H)-guai-9-ene, oplopanone, oxyphyllenodiol A, eudesm-3-ene-1,6-diol, rel1b,3a,6b-trihydroxyeudesm-4-ene, arteincultone. Flavonoids: 7,40 -O-dimethylscutellar-ein(5,6-dihydroxy-7,40 -dimethoxyflavone) and salvigenin. Glycosides: teucardoside and poliumoside.

44

T. polium L.

Europe

Ethanol

70%

Rutin, apigenin, apigenin-4, 7-dimethylether, cirsimaritin, cirsiliol, luteolin, 6-hydroxyluteolin, luteolin-7-O-glucoside, salvigenin, apigenin 7-glucoside, eupatorin, apigenin-5 galloylglucoside, 3′,6- dimethoxy apigenin, 4′,7- dimethoxy apigenin and salvigenin.

45

  • On the other hand, about the extraction procedure, time and temperature should be indicated as well as in other experiments like cytotoxic and antibacterial activity.

The methods used were described in details.

  • Extraction procedure

The fresh aerial flowering parts were dried at room temperature for seven to ten days. For the bioactive compounds extraction, 4g of the plant powder material were macerated in 40 mL absolute methanol (ratio: 1:10, w/v). Methanolic extracts were pooled, filtered, and the solvent was removed at 60â—¦C in the incubator chamber.

  • Evaluation of the cytotoxicity

The cytotoxicity of the T. polium extract was evaluated on VERO (African green monkey kidney) cell lines using the MTT colorimetric method. A decreasing concentration of the extract (from 3333 µg/mL to 1 µg/mL) diluted in MEM supplemented with 2%-FBS was applied on VERO cell in a 96-well plate. Cell controls were incubated under the same conditions without extracts. After 72 h of incubation at 37°C, the extract dilutions were substituted with the MTT solution (5 mg/mL), incubated for 2 h at 37°C, and then dissolved by the dimethyl sulfoxide (DMSO). The plate was read on an ELISA reader at a 570 nm wavelength to measure the optical density. The 50% cytotoxic concentration (CC50) was determined by regression analysis in comparison to negative control [83].  

  • Antibacterial and antifungal activity using disc diffusion assay

The disc diffusion assay was performed on Mueller-Hinton agar plates for all bacteria, Sabouraud chloramphenicol agar for yeasts, and Potato Dextrose agar for Aspergillus strains. 10 μL of methanolic extract mother solution (300 mg/ml) was used to impregnate 6 mm-sterile discs. The experiment was done in triplicate. All Petri dishes were incubated overnight at 37°C and the diameter of growth inhibition zone was recorded using 1 cm- flat ruler. The mean diameter was recorded and expressed as (GIZ mm±SD). Results obtained are interpreted using the scheme proposed by Parveen et al. [95]: no activity (GIZ 0), low activity (GIZ:1–6 mm), moderate activity (GIZ :7–10 mm), high activity (GIZ:11–15 mm), and very high activity (GIZ:16–20 mm). Ampicillin and Amphotericin B were used as control.

4.4.3. MICs and MBCs/MFCs determination using the microdilution assay

MICs and MBCs/MFCs values were determined by a microtitre broth dilution method as previously described by Snoussi et al. [96]. The test medium was Mueller–Hinton Broth (MHB) for bacterial strains, and Sabouraud dextrose broth for fungal isolates. Bacterial/fungal suspensions (100 μL) were inoculated into the wells of 96-well microtitre plates in the presence of samples with different final concentrations (ranging from 0.039 mg/ml to 100 mg/ml). The inoculated microplates were incubated at 37°C for 24 h. The lowest concentration of T. polium methanolic extract, which did not show any visual growth of tested organisms was recorded as MIC value (expressed in mg/mL). The MBCs/MFCs values are determined by streaking all wells after the MICs values on the correspondent agar media of the tested microorganisms. MBC/MIC ratio and MFC/MIC ratio were used to interpret the activity of the essential oil as described by Gatsing et al., [97].

  • Antiviral activity

The antiviral activity of T. polium methanolic extract was evaluated on Coxsakievirus B-3 (CVB3) and Herpes Simplex Virus type 2 (HSV-2). The viruses were propagated on Vero cells in MEM medium supplemented with 2% FBS at 37â—¦C under humidified 5% CO2 atmosphere. The infectious titer of the stock solution was 2x107 TCID50/mL (50% tissue culture infectious doses/mL) for CVB-3 and 8x105 PFU/mL (plaque formation units/mL, PFU) for HSV-2. A MOI (multiplicity of infection) of 0.01 for CVB-3 and 0.1 for HSV-2 was seeded into cells monolayers cultivated in 96-well culture plates (2x104 cells/wells), with different concentrations of the extract starting from the CC50 value. Plates without virus or extract were used as negative and positive controls, respectively. After 48 h incubation, the inhibition of the cytopathic effect (CPE) for CVB-3 and the plaque virus for HSV-2 was observed in an inverted microscope [94].

  • The method used to compound identification is missing.
  • Phytochemical analysis of polium extract.

The HR-LCMS analysis of T. polium was analyzed by using UHPLC-PDA-Detector Mass spectrometer (HR-LCMS 1290 Infinity UHPLC System, 1260 Infinity Nano HPLC with Chipcube, 6550 iFunnel QTOFs), Agilent Technologies, USA. For chromatographic separation, an Agilent 1200 Series HPLC system (Agilent Technologies, USA) equipped with a binary gradient solvent pump, HiP Sampler, column oven and MS Q-TOF with Dual AJS ES Ion Source. Samples were separated on SB-C18 column (2.1 × 50 mm, 1.8-particle size; Agilent Technologies, USA) maintained at 25°C. The solvents used were: water containing 0.1% HCOOH and Acetonitrile containing 10% water and 0.1% HCOOH. The following gradient elution program at a flow rate of 0.3 mL min−1 was applied. MS detection was performed in MS Q-TOF Mass spectrometer (Agilent Technologies). Compounds were identified via their mass spectra and their unique mass fragmentation patterns. Compound Discoverer 2.1, ChemSpider and PubChem were used as the main tools for the identification of the phytochemical constituents [93].

  • The abbreviations should be explain when mentioned the first time in the paper.

All abbreviations were explained in whole manuscript.

  • The use of VERO cell lines should be motivated.

VERO cell lines used for the antiviral activity because they are considered as permissive cell lines.

  • Results of cytotoxic effect should be better discussed and possibly correlated with other activities.

In fact, we discussed the result obtained with other published papers and with with chemical profile of T. polium extract.

Nematollahi-Mahani et al. [51] studied the toxicity of T. polium 96%-ethanolic extract on different known malignant cell lines like A549 (human lung adenocarcinoma), BT20 (human breast ductal carcinoma), MCF-7 (human breast adenocarcinoma), and PC12 (mouse pheochromocytoma). The results reported showed that IC50 values were 90 µg/ml (A549), 106 µg/ml (BT20), 140 µg/ml (MCF-7) and 120 µg/ml (PC12). The cytotoxic properties of T. polium aerial parts can be attributed the presence in the organic extracts of some diterpenoids and their acyl derivatives (teucvin and teucvidin), flavonoids (cirsiliol, cirsimaritin, cirsilineol, salvigenin and 5-hydroxy-6,7,3’,4’-tetramethoxyllavone), saponin poliusaposide [52], and selenium [53-55].

Round 2

Reviewer 1 Report

The novelty of the study in the context given by Authors is poor and unclear. Authors stated that (…) the main objectives of the present investigation were to assess for the first time the phytochemical profile of T. polium methanolic extract from Hail region (Saudi Arabia) using High resolution-Liquid  chromatography  coupled  to  mass  spectrometry (..)” – it is hardly to assess the novelty of the paper. New methodology? Or new site of the plant?

It is hardly to agree with statement: “assess for the first time the phytochemical profile” – because only 29 compounds (according to Tab. 1) were found in extracts and I think it is not enough to use the term “phytochemical profile”. Moreover, on chromatogram (Fig.1) only 12 peaks are shown (what with the others?).

If the compounds in Table are new, MS identification is not enough to confirm their structure.

The method is not new and it was used in plant investigation. Moreover the term - liquid  chromatography  coupled with high resolution mass  spectrometry is more adequate.

On what basis the Authors stated “ The  tested  extract  was  dominated  by  3R-hydroxy-9E,11Z-octadecadienoic  acid,  rhoifolin,  sericetin  diacetate,  dihydrosamidinand  cepharanthine”. It is difficult to agree with this on the basis of the chromatogram. To confirm it quantitative analysis should be done.

On chromatogram (fig.1) peak no. (6). – Valtratum has retention time about 20 min, while in table 1 it is 9,5 min.

Why were the other species (not related with present study e.g. T. scordium, T. stocksianum) given in Table 5?

There are still many errors, e. g. line 100: “in fat”?; line 102 “3R-hydroxy-9E” ? (in table is: 13R-hydroxy-9E…),  lack of capital letters, doubled phrase in Table 5: μg/mL  extract  μg/mL  extract), erroneous format of fonts e.g. in legend for Fig. 1……etc

The quality of Fig.1 is still very poor and the figure is unreadable (the same comment for Supplementary data)

In my opinion, it is only preliminary study. Lack of correlation of extract composition with activity.  I agree with Authors that “Further analysis are needed to correlate between the obtained results with the phytochemical composition of organic extract (…).”

Author Response

Point-by-point response to Reviewer 1:

Dear Dr.  (Reviewer 1): Thank you for your second round consideration of our manuscript. We found the comments helpful, and believe our revised manuscript represents a significant improvement over our initial submission.

Open Review

English language and style

( ) Extensive editing of English language and style required 
( ) Moderate English changes required 
( ) English language and style are fine/minor spell check required 
(x) I don't feel qualified to judge about the English language and style 

Yes

Can be improved

Must be improved

Not applicable

Does the introduction provide sufficient background and include all relevant references?

( )

( )

(x)

( )

Is the research design appropriate?

( )

( )

(x)

( )

Are the methods adequately described?

( )

( )

(x)

( )

Are the results clearly presented?

( )

( )

(x)

( )

Are the conclusions supported by the results?

( )

( )

(x)

( )

Comments and Suggestions for Authors

The novelty of the study in the context given by Authors is poor and unclear. Authors stated that (…) the main objectives of the present investigation were to assess for the first time the phytochemical profile of T. polium methanolic extract from Hail region (Saudi Arabia) using High resolution-Liquid  chromatography  coupled  to  mass  spectrometry (..)” – it is hardly to assess the novelty of the paper. New methodology? Or new site of the plant?

Special thanks for your remarks: In fact, the main goal of the present study was the identification of bioactives molecules from T. polium methanolic extract collected from Hail region (Saudia Arabia) using HR-LCMS technique.

It is hardly to agree with statement: “assess for the first time the phytochemical profile” – because only 29 compounds (according to Tab. 1) were found in extracts and I think it is not enough to use the term “phytochemical profile”. Moreover, on chromatogram (Fig.1) only 12 peaks are shown (what with the others?).

In fact, we added more details about the retention time, the chemical structure, area the pics obtained and chemical structure of the identified 29 compounds in the revised paper.

RT

Identified Compound Name

Chemical class

Chemical formula

1

0.945

10-Hydroxyloganin

Isoprenoid

C17H26O11

2

1.046

13R-hydroxy-9E,11Z octadecadienoic acid

Octadecanoid

C18H32O3

3

1.062

Bis (2-hydroxypropyl) amine

Amino alcohol

C6H15NO2

4

1.447

9-amino-nonanoic acid

Amino fatty acid

C9H19NO2

5

3.807

10-Amino-decanoic acid

Amino fatty acid

C10H21NO2

6

4.739

7-Epiloganin tetraacetate

Isoprenoid

C25H34O14

7

5.342

b-D-Glucopyranoside uronic acid, 6-(3-oxobutyl)-2- naphthalenyl

Organic acid, Phenol

C20H22O8

8

5.948

Cepharanthine

Alkaloid

C37H38N2O6

9

5.96

Troxerutin

Flavonol

C27H30O14

10

6.319

Deoxyloganin tetraacetate

Isoprenoid

C29H28O7

11

6.329

CMP-N-acetylneuraminic acid

Amino sugar

C33H42O19

12

6.376

Rhoifolin

Flavonoid

C25H34O13

13

7.838

Sericetin diacetate

Flavonol

C20H31N4O16P

14

8.198

Carapin-8 (9)-Ene

Limonoid

C27H30O7

15

8.848

Selinidin

Coumarin derivative

C19H20O5

16

9.1

Harpagoside

Iridoid glycoside

C24H30O11

17

9.126

8-Epiiridodial glucoside tetraacetate

Isoprenoid

C24H34O11

18

9.262

Larixol Acetate

Bicyclic labdane diterpenes

C22H36O3

19

9.525

Valtratum

Terpene

C22H30O8

20

9.807

Triptonide

Diterpene triepoxide

C20H22O6

21

10.036

Koparin 2'-Methyl Ether

Isoflavonoid

C17H14O6

22

10.779

Dihydrosamidin

Coumarins

C21H24O7

23

11.022

10S,11R-Epoxy-punaglandin 4

Eicosanoid

C25H35ClO9

24

11.279

16alpha,17beta-Estriol 3-(beta-D- glucuronide)

Steroidal glycosides

C24H32O9

25

11.28

16-Hydroxy-4-carboxyretinoic Acid

Isoprenoid

C20H24O5

26

12.149

Isotectorigenin, 7- Methyl ether

Isoflavonoid

C18H16O6

27

13.274

3-hydroxy-3',4'- Dimethoxyflavone

Flavonoid

C17H14O5

28

18.427

Khayanthone

Limonoid

C32H42O9

29

20.37

1-dodecanoyl-sn-glycerol

Glycerolipid

C14H22N2O3

Table 1. Bioactive compounds and their chemical class identified in T. polium methanolic extract using HR-LCMS technique.

In fat, T. polium methanolic extract was dominated by terpenes and their derivatives, fatty acids and their derivatives, polyphenols, flavonoids and derivatives, coumarines, amino acid derivatives, and alkaloids. The tested extract was dominated by 3R-hydroxy-9E,11Z-octadecadienoic acid, rhoifolin, sericetin diacetate, dihydrosamidin and cepharanthine. The chromatograms (figure 1) showed respectively major peaks indicating the presence of various phytochemical constituents. Data related to MS spectra, MS/MS spectrum peak list, chemical formula, and m/z values are listed in the supplementary file (Supplementary data S1).

Figure 1. Main identified phytocompounds in T. polium methanolic extract using HR-LCMS technique. (2A): +ve Chromatogram – (1). 13R-hydroxy-9E,11Z-octadecadienoic acid, (2). Rhoifolin, (3). Sericetin diacetate, (4). Selinidin, (5). Harpagoside, (6). Valtratum, (7). Triptonide, (8). Koparin 2'-Methyl Ether, (9). Dihydrosamidin, (10). 10S,11R-Epoxy-punaglandin, (11). 4, 16alpha, 17beta-Estriol 3-(beta-D-glucuronide), (12). Khayanthone. (2B): -ve Chromatogram – (1). 10-Hydroxyloganin, (2). 7-Epiloganin tetraacetate, (3). Cepharanthine, (4). Deoxyloganin tetraacetate, (5). Carapin-8 (9)-Ene, (6). 1-dodecanoyl-sn-glycerol.

The chemical structure of the identified 29 compounds are listed in Figure 2.

Figure 2. Chemical structure of the 29 compounds identified in T. polium methanolic extract byg HR-LCMS. The name of compounds (1-29) are listed in table 1.

If the compounds in Table are new, MS identification is not enough to confirm their structure.

In fact, the reported compounds were identified according to their spectrum peak list in both MS and MSn as reported in the Supplementary data. These compounds are not new identified ones, but are well known compounds found in diverse plant species. The importance of our work is that we report for the first time the identification of these components in T. polium methanolic extract from the aerial parts using HR-LCMS technique.

The method is not new and it was used in plant investigation.

 Previously, different techniques were used to identify and screen methanolic extracts from the same plant organs in different countries around the word. As summarized in table 5, the identified compounds are reported for the first time in this plant material (same plant, same organs, same protocol of extraction and same solvent used) but using another analytical technique (HR-LCMS).

Moreover the term - liquid  chromatography  coupled with high resolution mass  spectrometry is more adequate.

We also used the suggested term: liquid  chromatography  coupled with high resolution mass  spectrometry in the new revised version of our manuscript.

 On what basis the Authors stated “ The  tested  extract  was  dominated  by  3R-hydroxy-9E,11Z-octadecadienoic  acid,  rhoifolin,  sericetin  diacetate,  dihydrosamidinand  cepharanthine”. It is difficult to agree with this on the basis of the chromatogram. To confirm it quantitative analysis should be done.

In fact, the dominance was estimated according the area of pic obtained for each compound in the chromatograms obtained.

On chromatogram (fig.1) peak no. (6). – Valtratum has retention time about 20 min, while in table 1 it is 9,5 min.

Figure 1. Main identified phytocompounds in T. polium methanolic extract using HR-LCMS technique. (2A): +ve Chromatogram – (1). 13R-hydroxy-9E,11Z-octadecadienoic acid, (2). Rhoifolin, (3). Sericetin diacetate, (4). Selinidin, (5). Harpagoside, (6). Valtratum, …..

Why were the other species (not related with present study e.g. T. scordium, T. stocksianum) given in Table 5?

In fact, the cited Teucrium species other than polium species in Table 5 were used for the comparison and correlation between species. But we removed these non polium species from the table 5.

There are still many errors, e. g. line 100: “in fat”?; line 102 “3R-hydroxy-9E” ? (in table is: 13R-hydroxy-9E…),  lack of capital letters, doubled phrase in Table 5: μg/mL  extract  μg/mL  extract), erroneous format of fonts e.g. in legend for Fig. 1……etc

The whole manuscript was entirely revised for grammatical and spelling errors. We hope that the new submitted version of the manuscript is correct.

The quality of Fig.1 is still very poor and the figure is unreadable (the same comment for Supplementary data)

The figure 1 was entirely changed to increase its readability.

In my opinion, it is only preliminary study. Lack of correlation of extract composition with activity.  I agree with Authors that “Further analysis are needed to correlate between the obtained results with the phytochemical composition of organic extract (…).”

In fact, as we suggested the correlation between the composition and biological activity needs further studies and new experiments to can use results as proof for a possible correlation between the suggested phytochemical composition and all obtained biological properties of the tested extract. In fact, in silico approach is now running in order to correlate between composition/biological properties.

Finally, special thanks for your important comments made on our manuscript helping us to reorganize and format the presented results in the present study.

                    All the best

                                                                                                   Prof. Dr. Mejdi Snoussi

Reviewer 2 Report

The Authors clearly addressed all previous reviewer' comments.

Author Response

Dear colleague

Special thanks for your previous remarks about our manuscript

Special thanks for accepting our paper.

All the best

Round 3

Reviewer 1 Report

I accept the manuscript in present form